# Mixup Your Own Pairs: Supervised Contrastive Learning for Regression with Mixup

## Abstract

In representation learning, regression has traditionally received less attention than classification. Directly applying representation learning techniques designed for classification to regression often results in fragmented representations in the latent space, yielding sub-optimal performance. In this paper, we argue that the potential of contrastive learning for regression has been overshadowed due to the neglect of two crucial aspects: *ordinality-awareness* and *hardness*. To address these challenges, we advocate "mixup your own contrastive pairs for supervised contrastive regression", instead of relying solely on real/augmented samples. Specifically, we propose **Sup**ervised Contrastive Learning for **Re**gression with **Mix**up (**SupReMix**). It takes *anchor-inclusive* mixtures (mixup of the anchor and a distinct negative sample) as hard negative pairs and *anchor-exclusive* mixtures (mixup of two distinct negative samples) as hard positive pairs at the embedding level. This strategy formulates harder contrastive pairs by integrating richer ordinal information. Through extensive experiments on six regression datasets including 2D images, volumetric images, text, tabular data, and time-series signals, coupled with theoretical analysis, we demonstrate that **SupReMix** pre-training fosters continuous ordered representations of regression data, resulting in significant improvement in regression performance. Furthermore, SupReMix is superior to other approaches in a range of regression challenges including transfer learning, imbalanced training data, and scenarios with fewer training samples.

## 1 Introduction

Regression problems aim to predict continuous values based on given input data. They encompass a broad range of application domains such as age estimation based on human appearance (Rothe et al., 2018; Moschoglou et al., 2017), quantification of Semantic Textual Similarity (STS) (Cer et al., 2017), and prediction of health scores from physiological signals (Bookheimer et al., 2019; Peng et al., 2021). *Vanilla deep regression* refers to the approach of training a deep model to estimate the target value, with the distance (e.g., L1 distance) between the prediction and ground-truth defined as the loss function (Lathuilière et al., 2019). There are alternative approaches to formalize regression as classification tasks by discretizing the output space to multiple classes and training the model with cross-entropy loss (Niu et al., 2016).

Most prior research has concentrated on training models in an end-to-end manner, lack of exploration specifically for regression representation learning. In the realm of classification, techniques such as supervised contrastive learning (SupCon) (Khosla et al., 2020) have achieved significant success in enhancing representation accuracy. One might consider directly adapting SupCon for regression tasks. However, such direct application of SupCon tends to neglect the inherent ordinal nature of regression, i.e., lack of *ordinality-awareness*. This oversight is evident from the fragmented representations and the unchanged characterizations after label permutation, as illustrated in the top row of Figure 1. Furthermore, prior work in classification underscores the significance of hard contrastive pairs in contrastive learning (Ho & Nvasconcelos, 2020; Robinson et al., 2020; Kalantidis et al., 2020; Wu et al., 2023). Nevertheless, they mainly focused on hard negatives with hard positives underexplored, and the *hardness* of contrastive pairs in contrastive learning for regression remains inadequately examined. Data mixing techniques (Zhang et al., 2018; Verma et al., 2019; Shen et al., 2022), commonly used for data augmentation, have been used to create hard samples in previous contrastive learning methods for classification (Kalantidis et al., 2020; Lee et al., 2020; Liu et al., 2023). However, these methods do not leverage the label distance to differentiate the hardness among hard negative mixtures. One recent work using supervised contrastive regression (SupCR) (Zha et al., 2022) aims to refine continuous representations for regression. Nevertheless, its dependency on data augmentation restricts its applicability to domains where effective augmentation techniques have been established, such as time series or tabular data.

In this paper, we propose *Supervised Contrastive Learning for Regression with Mixup (SupReMix)*, a new framework for regression representation learning. Our aim is to better leverage the inherent ordinal relationships among various inputs and foster the generation of "harder" contrastive pairs. Instead of relying on real samples in conventional contrastive learning, the proposed SupReMix approach constructs new contrastive pairs at the

embedding level in an *anchor-inclusive* and *anchor-exclusive* manner. We take *anchor-inclusive* mixtures as hard negatives: mixing the anchor with a distinct negative sample, thus pulling the negatives closer to the convex hull between the anchor and negatives and encourage continuity. On the other hand, we take *anchor-exclusive* mixtures as hard positives: merging two negative samples, the convex combination of whose labels equals to that of the anchor, to encourage local linearity. Moreover, we assign weights to the negative pairs to incorporate label distance information. Our theoretical analysis offers a robust foundation, demonstrating that SupReMix is capable of leading to the formation of continuous ordered representations. For the remainder of the paper, we will refer to our hard negatives and hard positives as "Mix-neg" and "Mix-pos", respectively.

To validate our approach, we perform extensive experiments on six datasets that span across various domains/modalities. In all cases our SupReMix method consistently surpasses other supervised contrastive learning frameworks and vanilla deep regression, securing performance enhancements by **11%** on average across six datasets. Notably, in tackling the imbalanced regression scenario, SupReMix operates orthogonally with pre-existing methods to facilitate an over **30%** improvement compared to vanilla deep regression. Furthermore, our method showcases proficiency in transfer learning from large to small datasets, leading to a gain of **29%** compared to vanilla. Additionally, it also demonstrates resilience to the reduced size of the training set. These confirm the versatility and efficacy of SupReMix in a broad spectrum of regression problems.

## 2 UNDERSTANDING CONTRASTIVE LEARNING FOR REGRESSION

In this section, we show that direct application of contrastive learning for classification to regression tend to learn discontinuous representations that are insensitive to label permutation, and its swift saturation of training using conventional positive/negative contrastive pairs.

### 2.1 ORDINALITY-AWARENESS

In a well-constructed model for regression representation, data points ordered by their labels should also exhibit ordinality in the latent space. This implies that generated embeddings should be arranged in accordance with the label order. Conversely, when we apply label permutation to the regression dataset (ensuring that data with identical labels remain identical and those with different labels remain distinct) a model sensitive to ordinality should exhibit degraded performance. In the top row of Figure 1, we utilized the UCI-Airfoil dataset (Dua & Graff, 2019) to show that the previous approach SupCon produces embeddings that consistently display clustering patterns, regardless of whether the labels are genuine or permuted. In contrast, our proposed approach SupReMix produces continuous embeddings that reflect inherent ordered relationships when using genuine labels, but performance degrades with label permutation (see the bottom row of Figure 1) as expected.

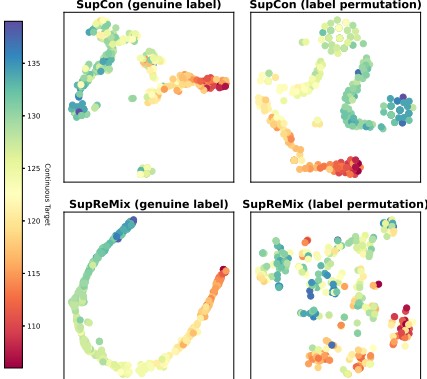

Figure 1: SupCon versus SupReMix: t-SNE (van der Maaten & Hinton, 2008) visualization of latent features on the UCI-airfoil test set.

### 2.2 HARDNESS

Previous studies on contrastive learning have demonstrated that as the training process advances, a diminishing number of hard negative samples make significant contributions to the overall loss function (Kalantidis et al., 2020). Our observations confirm this phenomenon and extend it to positive samples as well, particularly when applied to regression problems. In Figure 2, we utilized the AgeDB-DIR dataset (Moschoglou et al., 2017) to examine the distribution of logit values (i.e., the inner product of two embeddings with norm of 1) representing the similarity between two embeddings over the training process. We monitor all positive pairs and the top 1k most challenging negative pairs. We note that the logit values for positive pairs saturate swiftly, approaching 1 after around 100 training epochs. This indicates a reduced reliance on these pairs in later learning stages as they contribute less to the loss function. Similarly, fewer negative pairs contribute to the loss function over time.

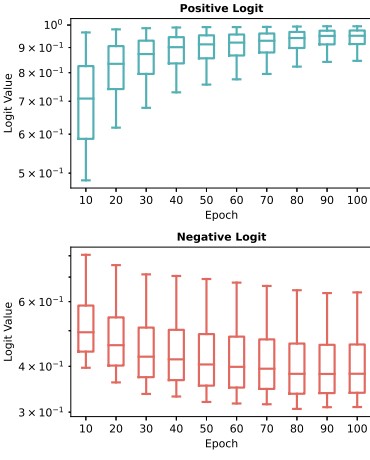

Figure 2: Logit distribution shift over training on the AgeDB-DIR dataset.

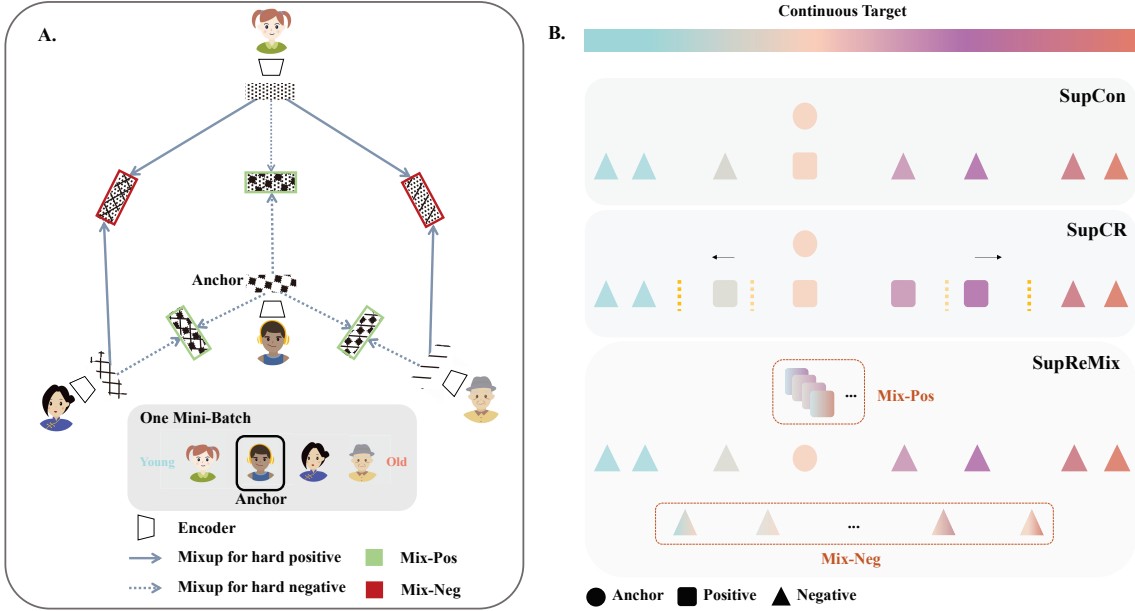

Figure 3: Schematic overview of SupReMix and comparison with SupCon and SupCR. **A.** An encoder first encodes inputs to embeddings. Given an anchor, Mix-neg are obtained through mixups ($\lambda_1 \sim \text{Beta}(\alpha,\beta)$) of the anchor itself and a negative in the latent space. Meanwhile, Mix-pos are derived from mixups ($\lambda_2$ is deterministic for two mixup embeddings) of two negative embeddings, the convex combination of whose labels equals to the anchor. **B.** SupCon identifies samples with the same label as positives and those with different labels as negatives, whereas SupCR determines positives and negatives through a relative approach. SupReMix further refines this process by introducing hard positives and negatives alongside the conventional real and augmented ones. SupReMix holds a key advantage over SupCR: it does not require data augmentation, which can be difficult when dealing with time series or tabular data.

## 3 SUPERVISED CONTRASTIVE LEARNING FOR REGRESSION WITH MIXUP

In a regression task, our goal is to train a neural network that consists of two main components: an encoder $f(\cdot)$: $X \mapsto \mathbf{z} \in \mathbb{R}^{d_e}$ which encodes inputs to embeddings, and a predictor $p(\cdot)$: $\mathbf{z} \in \mathbb{R}^{d_e} \mapsto m \in \mathbb{R}$ which outputs the target value $m \in \mathbb{R}$. Given one mini-batch, hard contrastive pairs are first created utilizing the mixup technique. Following this, we calculate our supervised contrastive regression loss, denoted as $\mathcal{L}_{\text{SupReMix}}$, based on both the real and our hard contrastive pairs. To predict the target value, $f(\cdot)$ is followed by $p(\cdot)$, trained by a regression loss (e.g., L1 loss).

In this section, we outline our approach to supervised contrastive regression. We begin in Section 3.1 with an explanation of our mixup strategy for generating hard negative and positive pairs. This is followed by Section 3.2, which is about our weights defined for contrastive pairs. This introduces *distance magnifying (DM)*, a behavior that is greatly advantageous in supervised contrastive regression, differentiating it from classification. In Section 3.3, we bring together the preceding elements to formulate our supervised contrastive regression loss, $\mathcal{L}_{\text{SupReMix}}$. In Section 3.4, we present a theoretical analysis of the distance magnifying property of our weights for negative pairs, as well as the ordinality-awareness of $\mathcal{L}_{\text{SupReMix}}$.

**Notations.** Let $I$: the set of embeddings from real samples with $N := |I|$ (i.e. mini-batch size), $M \subset \mathbb{R}$: the set of all initial labels, $\rho: I \mapsto M$: the function mapping an embedding to its label. Order $I$ such that $\rho$ is monotone. $I_m \subset I$: the set of embeddings with $\rho = m$ and $k_m := |I_m|$, which we take to be 0, if $m \notin M$. We give an order for elements in $I_m$, $(m,i)$, meaning the $i$-th embeddings in $I_m$.

### 3.1 MIXUP FOR HARD CONTRASTIVE PAIRS

***Anchor-inclusive* mixtures are hard negatives.** Given an anchor, a "mixed" negative—created through the convex combination of the anchor itself and a real negative—can be more challenging to differentiate compared to a real negative. This occurs in the latent space where the mixed negative is pulled closer to the anchor, thereby diminishing the discernible differences between the anchor and the hard negative. Given an anchor $\mathbf{z}_{m,i} \in I_m$, we generate a set of Mix-neg $\mathbf{z}_{m,i}^-$ defined by:

$$\mathbf{z}_{m,i}^{-} = \lambda_1 \cdot \mathbf{z}_{m,i} + (1-\lambda_1) \cdot \mathbf{z}', \text{ where } \mathbf{z}' \in I \backslash I_m, \lambda_1 \sim \text{Beta}(\alpha,\beta), \text{ and } \overline{m} = \lambda_1 \cdot m + (1-\lambda_1) \cdot m' \quad (1)$$

$$k_{m,i}^{-} := N - k_m \quad (2)$$

where $k_{m,i}^{-}$ is the number of Mix-neg generated for the anchor $z_{m,i}$, $\overline{m}$ is the label of Mix-neg, which is the convex combination of anchor's label $m$ and a distinct negative's label $m'$.

Different from vanilla mixup (Zhang et al., 2018), we use $\lambda_1$ as a "control" of hardness in our Mix-neg, modulating it through the adjustment of $\alpha$ and $\beta$ parameters that shape the Beta distribution from which $\lambda_1$ is sampled. If we choose $\alpha$ and $\beta$ to produce a skewed distribution where a majority of the values cluster close to one, the anchor $\mathbf{z}_{m,i}$ will almost surely predominate over the real negative $\mathbf{z}'$, thereby generating a harder negative. Conversely, if we choose $\alpha$ and $\beta$ so that $\lambda_1$ is drawn from a distribution that leans heavily towards zero, the anchor $\mathbf{z}_{m,i}$ has a relatively smaller share in the mixup, resulting in a reduction of the mixup hardness.

***Anchor-exclusive*** **mixtures are hard positives.** In contrastive learning, a common practice is to create positive pairs by augmenting an anchor to generate different views (Chen et al., 2020a); despite their visual differences, these augmented input retain their core identity, as they originate from a single input. However, relying solely on this method can limit the richness of the learned representations, as it overlooks the potential value of incorporating other similar objects that offer additional valuable perspectives (Wu et al., 2023). In addition, it is not applicable for domains with no appropriate data augmentation methods such as time series or tabular data. Finally, this approach does not take into account the underlying ordinal relationships among inputs and labels, which is particularly important in regression tasks.

To address these limitations, we mix two negative embeddings with labels above and below the anchor to serve as hard positives. This strategy not only preserves the natural order of the data but also expands the diversity of the positive pairs, creating a more stringent constraint that guides the learning process to a locally linear embedding space. Given an anchor $\mathbf{z}_{m,i}$, and a window size $\gamma \in \mathbb{Z}^+$, we create Mix-pos $\mathbf{z}_{m,i}^+$:

$$\mathbf{z}_{m,i}^+ = \lambda_2 \cdot \mathbf{z}_{m',i'} + (1-\lambda_2) \cdot \mathbf{z}_{m'',i''}, \text{ where } i' < i < i'', i-i', i''-i \leq \gamma, \text{ and } \lambda_2 \cdot m' + (1-\lambda_2) \cdot m'' = m \quad (3)$$

$$k_{m,i}^+ := \sum_{j=1}^{\gamma} k_{i-j} \cdot \sum_{l=1}^{\gamma} k_{i+l} \quad (4)$$

where $k_{m,i}^+$ is the number of Mix-pos for the anchor $\mathbf{z}_{m,i}$. $0 < \lambda_2 < 1$ is deterministic for each Mix-pos.

### 3.2 DISTANCE MAGNIFYING

In the loss function derived from contrastive pairs, we incorporate a vital parameter - the label distance information for each negative pair (including both real and mixup instances). Leveraging label distance as a metric facilitates a more fundamental approach to handling negative pairs. For each negative pair $\mathbf{z}_m$ and $\mathbf{z}_{\overline{m}}$ ($m \neq \overline{m}$), we define a weight $w_{m,\overline{m}}$ as:

$$w_{m,\overline{m}} = \frac{1 + |m - \overline{m}|}{m_{\max} - m_{\min}} \quad (5)$$

where $m_{\max}$ and $m_{\min}$ are the maximum and minimum of regression label respectively. (Note that in the implementation, all contrastive pairs will be multiplied by $w$ in the loss calculation. Adding 1 to the numerator ensures that $w$ remains non-zero and the same for all positive pairs). In our loss function (Section 3.3), the logit corresponding to each contrastive pair is modulated by multiplication with $w$.

In addition to explicitly encoding label distance information for negative pairs, another critical effect of this weight is to facilitate *distance magnifying (DM)*, a characteristic we argue is highly beneficial in supervised contrastive regression, distinguishing it from classification. Analogously, for a student taking an exam, it is fundamental to answer the easier questions correctly to secure a high score, and basic knowledge harnessed to tackle simpler

questions lays the groundwork to address more complex ones. We assert that a similar principle should govern contrastive learning in regression tasks. For example, in the context of an age prediction task based on face images, it is implausible for a model to accurately differentiate between ages 30 and 40 if it cannot discern the more pronounced differences between ages 30 and 80.

In Theorem 1, we establish a theoretical analysis that our devised weighting scheme for negative pairs accentuates the influence of the larger label distance. This essentially means that we increase the penalty for negative pairs that are farther apart as compared to those that are closer.

### 3.3 DEFINITION OF LOSS FUNCTION

**Extended Notations.** $\overline{I}$: the new set of training embeddings after mixup ($I \subset \overline{I}$). $\overline{M} \subset \mathbb{R}$: the new set of labels ($M \subset \overline{M}$). Given an anchor $\mathbf{z}_{m,i}$, $I_{(m,i),\overline{m}}$: the set of the anchor's contrastive embeddings with $\rho = \overline{m} \in \overline{M}$, $k_{(m,i),\overline{m}} := |I_{(m,i),\overline{m}}|$. All the embeddings are normalized to norm 1 in the latent space. Then we formulate our label-wise loss function as:

$$\mathcal{L}_{\text{SupReMix}} = \sum_{m \in M} \frac{-1}{k_m} \sum_{i=1}^{k_m} \sum_{\substack{j=1 \\ j \neq i}}^{k_{(m,i),m}} \log \frac{\exp(\langle \mathbf{z}_{m,i}, \mathbf{z}_{m,j} \rangle / \tau)}{\sum_{\overline{m} \in \overline{M}} \sum_{l=1}^{\prime k_{(m,i),\overline{m}}} w_{m,\overline{m}} \exp(\langle \mathbf{z}_{m,i}, \mathbf{z}_{\overline{m},l} \rangle / \tau)}. \tag{6}$$

where $\sum'$ is the summation running over all $I_{(m,i),\overline{m}}$ except for $(m,i) = (\overline{m}, l)$.

In Lemma 1, we establish a lower bound, denoted as $\mathcal{L}^*$, for $\mathcal{L}_{\text{SupReMix}}$. Subsequently, in Theorem 2, we demonstrate that this lower bound $\mathcal{L}^*$ represents the infimum. Furthermore, we elucidate that $\mathcal{L}^*$ can be closed if, and only if, the embeddings adhere to a globally ordered arrangement in accordance to their labels and maintain a locally linear behavior.

### 3.4 THEORETICAL ANALYSIS

**Theorem 1** (Distance Magnifying). *Given any two negative pairs (real or mixture), $s_{i,j}^{m,m'} := \langle \mathbf{z}_{m,i}, \mathbf{z}_{m',j} \rangle$, $s_{i,l}^{m,m''} := \langle \mathbf{z}_{m,i}, \mathbf{z}_{m'',l} \rangle$, where $m \neq m' \neq m''$, $|m' - m| > |m'' - m|$, we always have $\nabla_1 = \frac{\partial \mathcal{L}}{\partial s_{i,j}^{m,m'}} > 0$, $\nabla_2 = \frac{\partial \mathcal{L}}{\partial s_{i,l}^{m,m''}} > 0$, $\frac{\nabla_1}{\nabla_2}\big|_{\text{with } w} > \frac{\nabla_1}{\nabla_2}\big|_{\text{without } w}$ for $\mathcal{L}_{\text{SupReMix}}$.*

**Lemma 1** (Lower bound). *$\mathcal{L}_{\text{SupReMix}}$ has a lower bound $\mathcal{L}^*$.*

**Theorem 2** (Infimum). *The lower bound $\mathcal{L}^*$ is the infimum of $\mathcal{L}_{\text{SupReMix}}$. $\mathcal{L}_{\text{SupReMix}}$ is closed to its infimum if, and only if, the following are true:*

1. *all the real samples with the same label $m$ are embedded close to some vector $\mathbf{z}_m$;*

2. *all Mix-pos of an anchor $(m,i)$ have embeddings close to $\mathbf{z}_m$;*

3. *all the negatives (real and Mix-neg) of an anchor $(m,i)$ have $\mathbf{z}_{m,i} \cdot \mathbf{z}_{m',j}$ that are not equal to 1.*

*which means that the embeddings are **globally ordered and locally linear**.*

*Proof.* Refer to Appendix A. □

In Theorem 1, the key insight is that SupReMix amplifies the penalty for more distant negative pairs compared to closer ones by assigning specific weights to these pairs. Meanwhile, Lemma 1 and Theorem 2 collectively highlight that the SupReMix loss function not only possesses a tight bound, but also ensures both the ordinality and continuity of representations as it converges towards the infimum.

## 4 RESULTS

**Datasets.** We performed experiments on the following six datasets that span different domains and input modalities including text, 2D/3D image, tabular input, and time-series. Complete descriptions of each dataset are in Appendix B.1.

- **UCI-airfoil** (Dua & Graff, 2019): This dataset, available through the UCI Machine Learning Repository, comprises tabular data encapsulating the aerodynamic and acoustic test results of various airfoil blade sections. The task is to predict the Scaled Sound Pressure Level (SSPL) based on a five-dimensional input.
- **AgeDB-DIR** (Moschoglou et al., 2017): This dataset consists of face images paired with the corresponding ages. Following (Yang et al., 2021), we assembled an imbalanced training set of 12.2k images and balanced validation and test sets, each containing 2.1k images. The task is to estimate age from the facial images.
- **STS-B-DIR** (Cer et al., 2017; Wang et al., 2018): Sourced from the Semantic Textual Similarity Benchmark, this dataset includes sentence pairs extracted from news headlines and other mediums, annotated with a continuous similarity score ranging between 0 and 5, derived from the evaluations of several annotators. We followed Yang et al. (2021) to construct an imbalanced training set with 5.2k pairs and balanced validation and test sets, each with 1k pairs. The prediction task involves determining the similarity score from a given pair of sentences.
- **IMDB-WIKI** (Rothe et al., 2018): This dataset is a face image dataset. We crafted a training set incorporating 191.5k images and allocated 11k images for the validation and test sets each. Same as AgeDB-DIR, the task is age estimation from face images.
- **UK Biobank** (Allen et al., 2012): This is a large-scale biomedical database that collects information from UK participants. Utilizing T1-weighted volumetric MRI scans, we perform brain age estimation. The dataset, curated following the preprocessing guidelines in ALF (2018) and additional filtering criteria outlined in the Appendix B.1, encompasses a training set of 19.5k scans, with 2.3k scans each in the validation and test sets. The goal is to predict chronological age from T1 scans.
- **HCP-Aging** (Bookheimer et al., 2019): As a segment of the Human Connectome Project (HCP) aimed at investigating human neural pathways, this dataset involves time series data from resting-state fMRI. Following the processing pipeline delineated in Kong et al. (2019); Schaefer et al. (2018), we extracted a training set from 456 participants and validation and test sets from 100 participants each. The objective is to predict chronological age using time series inputs.

**Baseline methods.** We evaluated SupReMix against benchmarks including SupCon (Khosla et al., 2020) and vanilla deep regression (Vanilla) across all datasets. Additionally, we performed comparisons with SupCR (Zha et al., 2022) [1] on all image datasets, and supervised SimCSE (Gao et al., 2021) on the text (STS-B-DIR) dataset. For 2D image datasets, we follow standard data augmentation (Chen et al., 2020a) in contrastive learning. For the 3D volumetric image dataset, we adopt the augmentation used in Taleb et al. (2020) for all baseline methods. For other datasets where augmentation methods are not clearly defined (i.e., Tabular, Text, and Time-Series), we do not apply any augmentation during training. Positive pairs are two (real/augmented) samples within one minibatch that share the same target value. More details on augmentation methods are in Appendix B.1 and the *discretization* alternative is compared in Appendix D.4.

**Backbone Networks.** For the 2D image datasets (AgeDB-DIR and IMDB-WIKI), we adopt a ResNet-50 (He et al., 2016) as the backbone network. Following Wang et al. (2018); Yang et al. (2021), we adopt BiLSTM + GloVe word embeddings baseline for STS-B-DIR. For UK Biobank, we adopt a 3D-ResNet-18 (Hara et al., 2018) as the backbone network. For HCP-Aging, we adopt an 18-layer 1D-ResNet (Hong et al., 2020) as the backbone network to process time-series data. For UCI-airfoil we adopt a simple MLP (Cheng et al., 2023) with three hidden layers (d-20-30-10-1).

**Evaluation Schemes.** To evaluate our model for supervised contrastive regression, for UCI-airfoil, AgeDB-DIR, STS-B-DIR and IMDB-WIKI, we follow the *linear probe protocol* (Chen et al., 2020a), freezing the backbone network $f(\cdot)$ after the pre-training and training solely a predictor $p(\cdot)$ on top of it. For UK Biobank and HCP-Aging, we *finetune* the whole network after pre-training.

**Evaluation Metrics.** We use common metrics for regression to evaluate the performance, including mean-absolute-error (MAE), mean-square-error (MSE), Geometric Mean (GM) error, and Pearson correlation coefficient ($\rho$).

**Implementation Details.** Across all experiments, we utilize the Adam optimizer (Kingma & Ba, 2014) with an initial learning rate of $10^{-3}$. For contrastive learning frameworks, all methods are trained for the same number of

---

[1] we re-implement SupCR due to the lack of code release

Table 1: Evaluation on AgeDB-DIR (2D image)

| Metrics | MAE ↓ | MSE ↓ | GM ↓ |
|---|---|---|---|
| Vanilla | 7.77 | 101.60 | 5.05 |
| SupCon (Khosla et al., 2020) | 8.23 | 113.01 | 5.31 |
| SupCR (Zha et al., 2022) | 7.86 | 102.84 | 5.15 |
| C-MixUp (Yao et al., 2022) | 7.66 | 99.82 | 4.82 |
| **SupReMix** | **7.12** | **87.65** | **4.58** |
| GAINS (**Ours** VS. Vanilla(%)) | **+9.1** | **+15.9** | **+10.3** |

Table 2: Evaluation on STS-B-DIR (Text)

| Metrics | $\rho$↑ | MSE ↓ | MAE ↓ |
|---|---|---|---|
| Vanilla | 0.744 | 0.974 | 0.794 |
| SupCon (Khosla et al., 2020) | 0.745 | 0.954 | 0.786 |
| SimCSE (Gao et al., 2021) | 0.756 | 0.935 | 0.776 |
| RoBERTa (Liu et al., 2019) | 0.751 | 0.987 | 0.777 |
| **SupReMix** | **0.760** | **0.893** | **0.759** |
| GAINS (**Ours** VS. Vanilla(%)) | **+2.2** | **+9.1** | **+4.6** |

Table 3: Evaluation on IMDB-WIKI (2D image)

| Metrics | MAE ↓ | MSE ↓ | GM ↓ |
|---|---|---|---|
| Vanilla | 5.55 | 65.24 | 3.30 |
| SupCon (Khosla et al., 2020) | 6.06 | 71.92 | 3.67 |
| SupCR (Zha et al., 2022) | 5.70 | 66.00 | 3.45 |
| C-MixUp (Yao et al., 2022) | 5.43 | 64.34 | 3.33 |
| **SupReMix** | **5.38** | **62.40** | **3.15** |
| GAINS (**Ours** VS. Vanilla(%)) | **+3.2** | **+4.6** | **+4.8** |

Table 4: Evaluation on UK Biobank (3D image)

| Metrics | $\rho$↑ | MAE ↓ | MSE ↓ |
|---|---|---|---|
| Vanilla | 0.834 | 3.11 | 15.31 |
| SupCon (Khosla et al., 2020) | 0.844 | 3.14 | 15.78 |
| SupCR (Zha et al., 2022) | 0.856 | 3.03 | 14.69 |
| C-MixUp (Yao et al., 2022) | 0.859 | 3.00 | 14.59 |
| **SupReMix** | **0.863** | **2.97** | **14.37** |
| GAINS (**Ours** VS. Vanilla(%)) | **+3.5** | **+4.7** | **+6.5** |

Table 5: Evaluation on UCI-airfoil (Tabular data)

| Metrics | MAE ↓ | MSE ↓ | GM ↓ |
|---|---|---|---|
| Vanilla | 6.56 | 69.32 | 8.33 |
| SupCon (Khosla et al., 2020) | 5.68 | 48.90 | 6.99 |
| **SupReMix** | **4.88** | **39.70** | **6.30** |
| GAINS (**Ours** VS. Vanilla(%)) | **+34.4** | **+74.6** | **+32.2** |

Table 6: Evaluation on HCP-Aging (Time series)

| Metrics | $\rho$↑ | MAE ↓ | MSE ↓ |
|---|---|---|---|
| Vanilla | 0.654 | 8.99 | 126.43 |
| SupCon (Khosla et al., 2020) | 0.592 | 9.81 | 160.93 |
| **SupReMix** | **0.744** | **8.43** | **116.11** |
| GAINS (**Ours** VS. Vanilla(%)) | **+13.8** | **+6.6** | **+8.9** |

epochs for a fair comparison, followed by either a linear probe or fine-tuning approach for 100 epochs. Temperature parameter ($\tau$) selections are dataset-dependent, with values of 0.3 for IMDB-WIKI, 0.5 for AgeDB-DIR, UK Biobank and HCP-Aging; and 1.0 for STS-B-DIR and UCI-airfoil. When applying a vanilla regression approach, our implementation follows the guidelines established in previous studies (Yang et al., 2021; Peng et al., 2021; Cheng et al., 2023). Complete implementation details are in Appendix C.2.

### 4.1 MAIN RESULT

Across all the six datasets with a variety of input data modalities (see Tables 1, 2, 3, 4, 5, 6), our proposed SupReMix method consistently outperformed all baseline methods. This underscores the robustness and superiority of the learned representations through SupReMix, paving the way for advanced deep regression applications in diverse contexts.

### 4.2 IMBALANCED REGRESSION

As demonstrated in Table 7, our SupReMix method offers a substantial improvement in addressing imbalanced regression issues, acting orthogonally to the established solutions represented by Yang et al. (2021). Our method boosts overall performance by an impressive margin of approximately **15%**. More notably, it increases the median and few-shot learning scores by **30%** when compared to results achieved through vanilla deep regression.

Table 7: Evaluation on imbalanced regression using AgeDB-DIR. Many: many-shot region (bins with >100 training samples), Few: few-shot region (bins with <20 training samples).

| Metrics | MAE ↓ | | | | GM ↓ | | | |
|---|---|---|---|---|---|---|---|---|
| Shot | All | Many | Med | Few | All | Many | Med | Few |
| Vanilla | 7.77 | 6.62 | 9.55 | 13.67 | 5.05 | 4.23 | 7.01 | 10.75 |
| LDS + FDS (Yang et al., 2021) | 7.55 | 7.01 | 8.24 | 10.79 | 4.72 | 4.36 | 5.45 | 6.79 |
| SupReMix + LDS | 6.78 | **5.97** | 8.08 | 10.80 | 4.27 | **3.69** | 5.70 | 7.78 |
| SupReMix + FDS | 6.75 | 6.04 | 7.86 | **10.42** | 4.29 | 3.78 | 5.33 | 7.78 |
| SupReMix + LDS + FDS | **6.73** | 6.03 | **7.71** | 10.65 | **4.27** | 3.77 | **5.27** | **7.61** |
| GAINS (**Best** VS. Vanilla (%)) | **+15.5** | **+10.9** | **+29.1** | **+31.2** | **+18.3** | **+14.6** | **+33.0** | **+41.3** |

### 4.3 TRANSFER LEARNING

We use IMDB-WIKI and AgeDB-DIR to examine the transfer learning capabilities across datasets for the same prediction task. The focus of this investigation is on the pre-training efficacy of the feature encoder $f(\cdot)$

Table 8: Evaluation on transfer learning

| Metrics | IMDB → AgeDB | | AgeDB → IMDB | |
|---|---|---|---|---|
| | MAE ↓ | GM ↓ | MAE ↓ | GM ↓ |
| Vanilla | 7.89 | 5.01 | 8.48 | 5.46 |
| SupCR | 6.60 | 4.37 | 8.40 | 5.44 |
| **SupReMix** | **6.11** | **3.86** | **8.14** | **5.19** |
| GAINS (**Ours** VS. Vanilla(%)) | **+29.1** | **+29.8** | **+4.2** | **+5.2** |

across different datasets, leveraging linear probe whereby only the predictor $p(\cdot)$ is trained on the target dataset. In the scenario where IMDB-WIKI serves as the source dataset and AgeDB-DIR acts as the target dataset, SupReMix shows a pronounced enhancement in performance. It leads to a reduction in MAE by a substantial margin of **29.1%**. This improvement is indicative of SupReMix's superior capacity to generalize features learned from IMDB-WIKI to AgeDB-DIR. On the other hand, when the source and target datasets are swapped, SupReMix still manages to enhance the performance. The improvement in MAE in this case is 4.2%.

## 4.4 RESILIENCE TO REDUCED DATA

In numerous real-world scenarios, assembling a large training set proves to be impractical due to the substantial time and expense involved in labeling. Consequently, enhancing the model's adaptability to limited training data becomes a desirable objective. We evaluate the data efficiency of SupReMix and Vanilla deep regression utilizing training sets of various sizes drawn from the IMDB-WIKI dataset, ranging from the full size of 191.5k samples to a modest one of 4k samples. As shown in Fig 4, SupReMix consistently outperforms Vanilla across all training set sizes. The performance (MAE) gap increases from 0.17 to 1.24 as the training set size reduces. This underlines SupReMix's outstanding data efficiency, emerging as particularly advantageous in scenarios constrained by limited training data.

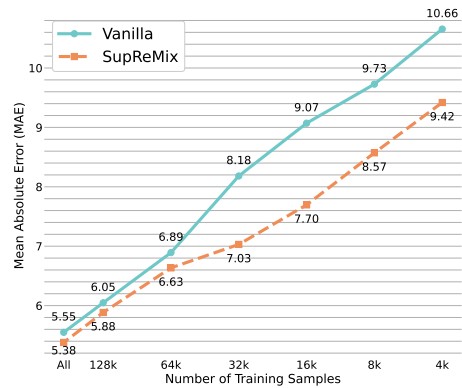

Figure 4: Evaluation on reduced training set using IMDB-WIKI

## 4.5 ABLATIONS AND HYPER-PARAMETERS

**Effectiveness of each proposed component.** To dissect the sources of SupReMix's performance gains, we isolate and examine three core components: Weight, Mix-neg, and Mix-pos. The "Weight" refers to the introduction of weights for negative pairs defined by label distance. Notably, the removal of all three components reverts SupReMix back to SupCon baseline.

As indicated in Table 9, removing any one of these components individually leads to a degradation in performance, thereby underscoring the effectiveness of each element in our proposed method. This ablation study unequivocally affirms the significant contributions of these components to the overall success of the SupReMix approach.

Table 9: Ablation study

| Metrics (AgeDB) | MAE ↓ | MSE ↓ | GM ↓ |
|---|---|---|---|
| SupCon | 8.23 | 113.01 | 5.31 |
| SupCon + Weight | 8.06 | 108.82 | 5.23 |
| SupCon + Mix-neg | 7.98 | 105.23 | 5.25 |
| SupCon + Mix-neg + Weight | 7.45 | 93.08 | 4.88 |
| SupReMix | 7.12 | 87.65 | 4.58 |

Table 10: Effect of $\gamma$

| AgeDB | MAE ↓ | STS-B | MAE ↓ |
|---|---|---|---|
| $\gamma=1$ | 7.36 | $\gamma=0.2$ | 0.78 |
| $\gamma=3$ | 7.28 | $\gamma=0.5$ | 0.76 |
| $\gamma=5$ | **7.12** | $\gamma=1.0$ | **0.76** |
| $\gamma=10$ | 7.55 | $\gamma=2.0$ | 0.79 |
| $\gamma=\infty$ | 8.00 | $\gamma=\infty$ | 0.79 |

**Choice of $\gamma$.** We introduce hyper-parameter $\gamma$ (Section 3.1) as the window size for Mix-pos. When we increase $\gamma$, samples that are more distant from the anchor will be used to mixup hard positive pairs. From Table 10, we can see that a moderate window size is optimal. A small window size limits the number of Mix-pos while a whole-range window size ($\gamma=\infty$) implicitly asserts the global linearity of the representation, which introduces noise.

**Choice of beta distribution.** To modulate the hardness for Mix-neg, we adjust the shape of the Beta distribution used for sampling $\lambda_1$. We investigate three different distributions: Beta(2,8), Beta(8,2), and Beta(5,5), each corresponding to different 'hardness' settings for the $\lambda_1$ sampling process. Using the Beta(2,8) distribution, characterized as the 'easy' mode, the anchor weighs less in the mixture with higher probability, facilitating a gentler introduction of hard negative pairs. In contrast, the 'hard' mode is represented by the Beta(8,2) distribution, which leads to a generation of more challenging negative pairs by attributing a larger share to the anchor. The Beta(5,5) distribution offers a balanced, or 'median' mode with a symmetric sampling strategy. As illustrated in Table 11, the Beta(2,8) distribution consistently surpasses the other sampling strategies in performance. This indicates that an excessively challenging generation of hard negative pairs can potentially impair the quality of the learned representation. By maintaining a more moderate difficulty level for Mix-neg, the 'easy' mode ensures a more productive learning process.

Table 11: Choice of beta distribution. Here we show the effect of varying the beta distribution on AgeDB-DIR (MAE / GM), STS-B-DIR (MSE / $\rho$), IMDB-WIKI (MAE / GM), UK Biobank (MAE / MSE), UCI-airfoil (MAE / GM) and HCP-Aging (MAE / GM)

| $(\alpha,\beta)$ | AgeDB | STS | IMDB | UKB | UCI | HCP |
|---|---|---|---|---|---|---|
| Beta($\alpha$=2, $\beta$=8) | **7.12** | **0.89** | **5.38** | **2.98** | **4.88** | **8.43** |
| | **4.58** | **0.76** | **3.15** | **14.37** | **6.30** | **5.47** |
| Beta($\alpha$=8, $\beta$=2) | 7.65 | 0.93 | 5.43 | 3.07 | 5.15 | 8.57 |
| | 4.82 | 0.75 | 3.18 | 15.26 | 6.41 | 5.57 |
| Beta($\alpha$=5, $\beta$=5) | 7.56 | 0.90 | 5.51 | 3.08 | 5.22 | 8.77 |
| | 4.86 | **0.76** | 3.26 | 15.33 | 6.47 | **5.47** |

## 5    RELATED WORK

**Contrastive Learning.** Contrastive learning has emerged as a powerful strategy in self-supervised learning, demonstrating improved performance through the alignment of positive pairs and the repulsion of negative pairs in a representation space (Chen et al., 2020a; He et al., 2020; Chen et al., 2020b). Its supervised variant, termed supervised contrastive learning (SupCon), has been devised as a generalization of triplet (Weinberger & Saul, 2009) and N-pair losses (Sohn, 2016), wherein pairs of samples from identical classes are considered positive pairs, and those from different classes are seen as negative pairs (Khosla et al., 2020). Recently, there have been various adaptations of contrastive learning to continuous labels, with each bringing distinct perspectives to the table. Dufumier et al. (2021) leveraged contrastive loss adjusted by continuous metadata for classification, while Schneider et al. (2023b) utilized a refined contrastive loss to encode behavioral and neural data through interpretable embeddings derived from continuous or temporal labels. Notably, unlike our approach, these studies do not explore regression problems. Yu et al. (2021) devised an action quality assessment model using score regression between two videos, bypassing the usual contrastive learning framework. Wang et al. (2022) added a contrastive loss term to the L1 loss to aid gaze estimation domain adaptation, improving cross-domain performance but reducing source dataset efficiency. In contrast, our method enhances the performance of the source dataset while at the same time facilitating domain adaptation. Zha et al. (2022) proposed supervised contrastive regression (SupCR), an improved contrastive loss defining positive and negative pairs in a relative way, which refines continuous representations for regression. However, its reliance on data augmentation limits its utility in areas devoid of robust augmentation methods, such as time series or tabular data. Contrarily, our approach remains applicable to those domains without necessitating augmentation.

**Hard Contrastive Pairs.** Prior research in classification emphasizes the crucial role of hard contrastive pairs in contrastive learning. It generally falls into two categories: hard sample mining and hard sample generation. The former aims to identify the most challenging samples from existing ones, with notable work by Robinson et al. (2020) devising an importance-based sampling strategy for mining hard negatives without computational overhead. In the realm of hard sample generation, Ho & Nvasconcelos (2020) utilized adversarial attacks to augment the training dataset, incorporating pixel-level disturbances into clean samples. Kalantidis et al. (2020) proposed hard negative mixing strategies at the feature level. Wu et al. (2023) developed a data generation framework enhancing contrastive learning through combined hard sample creation and contrastive learning. These approaches markedly diverge from ours, mainly as they revolve around self-supervised contrastive learning without utilizing label information, and target classification rather than regression. Yao et al. (2022); Schneider et al. (2023a) proposed data augmentation methods for regression tasks. However, applying them to contrastive learning frameworks has not been fully explored.

## 6    CONCLUSION

In this paper, we propose *Supervised Contrastive Learning for Regression with Mixup (SupReMix)*, a novel framework that generates hard negatives and hard positives for supervised contrastive regression in an *anchor-inclusive* and *anchor-exclusive* manner respectively on the embedding level. Supported by theoretical analysis, SupReMix leads to continuous ordered representations for regression. Extensive experiments have shown that SupReMix consistently improves over baselines, including vanilla deep regression counterparts and other supervised contrastive learning frameworks, across datasets, tasks, and input modalities. Moreover, we show the representations learned by SupReMix boost the performance for imbalanced regression, transfer learning and are more resilient to reduced training data.

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

# A  PROOFS

In this appendix, $z_{m,i}$ is used to denote normalized embeddings, whereas $z_{m,i}^*$ is used to denote unnormalized embeddings, in other words, we have $z_{m,i} = \frac{z_{m,i}^*}{||z_{m,i}||}$.

**Theorem 1** (Distance Magnifying). *Given any two negative pairs (real or mixture), $s_{i,j}^{m,m'} := \langle \mathbf{z}_{m,i}, \mathbf{z}_{m',j} \rangle$, $s_{i,l}^{m,m''} := \langle \mathbf{z}_{m,i}, \mathbf{z}_{m'',l} \rangle$, where $m \neq m' \neq m''$, $|m' - m| > |m'' - m|$, we always have $\nabla_1 = \frac{\partial \mathcal{L}}{\partial s_{i,j}^{m,m'}} > 0$, $\nabla_2 = \frac{\partial \mathcal{L}}{\partial s_{i,l}^{m,m''}} > 0$, $\frac{\nabla_1}{\nabla_2}\big|_{\text{with } w} > \frac{\nabla_1}{\nabla_2}\big|_{\text{without } w}$ for $\mathcal{L}_{SupReMix}$.*

*Proof.* For $m \neq m'$, we have

$$\frac{\partial \mathcal{L}}{\partial s_{i,j}^{m,m'}} = \frac{b_{m,i} \cdot w_{m,m'} \exp(\langle z_{m,i}, z_{m',j} \rangle / \tau)}{C_{m,i}^w \cdot \tau} > 0$$

where

$$C_{m,i}^w := \sum_{n \in \overline{M}} \sum_{l=1}^{k_{(m,i),n}}{}' w_{m,n} \exp(\langle z_{m,i}, z_{n,l} \rangle / \tau), \quad b_{m,i} = \frac{k_{(m,i),m} - 1}{k_m}.$$

Consider $w_{m,m'} = \frac{1 + t|m - m'|}{m_{\max} - m_{\min}}$. For comparison, we have

$$
\begin{aligned}
\frac{\frac{\partial \mathcal{L}}{\partial s_{i,j}^{m,m'}}}{\frac{\partial \mathcal{L}}{\partial s_{i,l}^{m,m''}}} &= \frac{b_{m,i} \cdot w_{m,m'} \exp(\langle z_{m,i}, z_{m',j} \rangle / \tau)}{C_{m,i}^w \cdot \tau} \cdot \frac{C_{m,i}^w \cdot \tau}{b_{m,i} \cdot w_{m,m''} \exp(\langle z_{m,i}, z_{m'',l} \rangle / \tau)} \\
&= \frac{w_{m,m'}}{w_{m,m''}} \cdot \exp(z_{m,i} \cdot (z_{m',j} - z_{m'',l}) / \tau) \\
&= \frac{1 + t|m - m'|}{1 + t|m - m''|} \cdot \exp(z_{m,i} \cdot (z_{m',j} - z_{m'',l}) / \tau),
\end{aligned}
\tag{7}
$$

Then we have

$$\frac{\frac{\partial \mathcal{L}}{\partial s_{i,j}^{m,m'}}}{\frac{\partial \mathcal{L}}{\partial s_{i,l}^{m,m''}}}\Bigg|_{t=1} > \frac{\frac{\partial \mathcal{L}}{\partial s_{i,j}^{m,m'}}}{\frac{\partial \mathcal{L}}{\partial s_{i,l}^{m,m''}}}\Bigg|_{t=0}$$

□

**Lemma 1** (Lower bound). *$\mathcal{L}_{SupReMix}$ has a lower bound $\mathcal{L}^*$.*

*Proof.* Recall we have

$$\mathcal{L}_{\text{SupReMix}} = \sum_{m \in M} \frac{1}{k_m} \sum_{i=1}^{k_m} \sum_{\substack{j=1 \\ j \neq i}}^{k_{(m,i),m}} \log \frac{\sum_{\overline{m} \in \overline{M}} \sum_{l=1}^{k_{(m,i),\overline{m}}}{}' w_{m,\overline{m}} \exp(\langle z_{m,i}, z_{\overline{m},l} \rangle / \tau)}{\langle \exp(z_{m,i}, z_{m,j}) / \tau \rangle}$$

Since the logarithmic function is monotone, and both weight and exponential function are positive, we keep only the positive pairs in the numerators and have

$$\mathcal{L}_{\text{SupReMix}} \geq \sum_{m \in M} \frac{1}{k_m} \sum_{i=1}^{k_m} \sum_{\substack{j=1 \\ j \neq i}}^{k_{(m,i),m}} \log \frac{\sum_{\substack{l=1 \\ l \neq i}}^{k_{(m,i),m}} \frac{1}{m_{\max}-m_{\min}} \exp(\langle z_{m,i}, z_{m,l}\rangle/\tau)}{\exp(\langle z_{m,i}, z_{m,j}\rangle/\tau)}$$

$$= -\sum_{m \in M} \frac{1}{k_m} \sum_{i=1}^{k_m} \sum_{\substack{j=1 \\ j \neq i}}^{k_{(m,i),m}} \log \frac{\exp(\langle z_{m,i}, z_{m,j}\rangle/\tau)}{\sum_{\substack{l=1 \\ l \neq i}}^{k_{(m,i),m}} \frac{1}{m_{\max}-m_{\min}} \exp(\langle z_{m,i}, z_{m,l}\rangle/\tau)} \tag{8}$$

$$= -\sum_{m \in M} \frac{1}{k_m} \sum_{i=1}^{k_m} \sum_{\substack{j=1 \\ j \neq i}}^{k_{(m,i),m}} \log \frac{\frac{1}{m_{\max}-m_{\min}} \exp(\langle z_{m,i}, z_{m,j}\rangle/\tau)}{\sum_{\substack{l=1 \\ l \neq i}}^{k_{(m,i),m}} \frac{1}{m_{\max}-m_{\min}} \exp(\langle z_{m,i}, z_{m,l}\rangle/\tau)} + C,$$

where $C = -\log(m_{\max} - m_{\min}) \cdot \sum_{m \in M} \frac{1}{k_m} \sum_{i=1}^{k_m} (k_{(m,i),m} - 1)$ is a constant. Now since $-\log$ is convex, by Jensen's inequality, we have

$$\mathcal{L}_{\text{SupReMix}} = -\sum_{m \in M} \frac{1}{k_m} \sum_{i=1}^{k_m} (k_{(m,i),m}-1) \sum_{\substack{j=1 \\ j \neq i}}^{k_{(m,i,m)}} \frac{1}{(k_{(m,i),m}-1)} \log \frac{\frac{1}{m_{\max}-m_{\min}} \exp(\langle z_{m,i}, z_{m,j}\rangle/\tau)}{\sum_{\substack{l=1 \\ l \neq i}}^{k_{(m,i),m}} \frac{1}{m_{\max}-m_{\min}} \exp(\langle z_{m,i}, z_{m,l}\rangle/\tau)} + C$$

$$\geq -\sum_{m \in M} \frac{1}{k_m} \sum_{i=1}^{k_m} (k_{(m,i),m}-1) \log \left( \frac{\sum_{\substack{l=1 \\ l \neq i}}^{k_{(m,i),m}} \frac{1}{m_{\max}-m_{\min}} \exp(\langle z_{m,i}, z_{m,l}\rangle/\tau)}{(k_{(m,i),m}-1) \sum_{\substack{l=1 \\ l \neq i}}^{k_{(m,i),m}} \frac{1}{m_{\max}-m_{\min}} \exp(\langle z_{m,i}, z_{m,l}\rangle/\tau)} \right) + C$$

$$= \sum_{m \in M} \sum_{i=1}^{k_m} \frac{(k_{(m,i),m}-1) \log(k_{(m,i),m}-1)}{k_m} + C$$

$$= \sum_{m \in M} \frac{1}{k_m} \sum_{i=1}^{k_m} (k_{(m,i),m}-1) \log \frac{k_{(m,i),m}-1}{m_{\max}-m_{\min}} =: \mathcal{L}^* \tag{9}$$

$\square$

**Theorem 2** (Infimum). *The lower bound $\mathcal{L}^*$ is the infimum of $\mathcal{L}_{\text{SupReMix}}$. $\mathcal{L}_{\text{SupReMix}}$ is closed to its infimum if, and only if, the following are true:*

1. *all the real samples with the same label $m$ are embedded close to some vector $\mathbf{z}_m$;*

2. *all Mix-pos of an anchor $(m,i)$ have embeddings close to $\mathbf{z}_m$;*

3. *all the negatives (real and Mix-neg) of an anchor $(m,i)$ have $\mathbf{z}_{m,i} \cdot \mathbf{z}_{m',j}$ that are not equal to $1$.*

*which means that the embeddings are **globally ordered and locally linear**.*

*Proof.* Let $\epsilon, \delta > 0$ be given. We aim to demonstrate the existence of a positive real number $\tau_0$ such that for all $\tau > \tau_0$, embeddings with $\mathcal{L} < \mathcal{L}^* + \epsilon$ can be obtained with a probability exceeding $1 - \delta$.

Firstly, we assign identical embedding $z_m$ to all samples labeled with $m$ belonging to the set $M$. Subsequently, we set these embeddings, denoted as $z_m$, to lie on a common plane. Without loss of generality, we can assume this plane to be spanned by the first two coordinate axes, thereby effectively reducing the embedding space to two dimensions.

Consider $z_m^*$ as the anchor embedding before normalization. We further take the $z_m^*$ to lie on a line such that their position on the line is proportional to their label, i.e.,

$$\frac{m - m'}{m - m''} = \frac{||z_m^* - z_{m'}^*||}{||z_m^* - z_{m''}^*||}, \quad \forall m \neq m' \neq m'' \in M$$

then the Mix-pos associated with label $m$ will share the same embedding, $z_m$. When it comes to Mix-neg, note that for any sample $(m,i)$, only a finite number of Mix-neg exist. We define an event $E_{m,i}$ to occur if at least one Mix-neg of $(m,i)$ forms an angle smaller than $\theta_{m,i}$ with the anchor $(m,i)$. We choose $\theta_{m,i}$ such that the event $E_{m,i}$ has a probability less than $\frac{\delta}{N}$. This ensures that the cumulative probability of all $E_{m,i}$ events being false surpasses $1-\delta$. Define $\theta_0$ to be the minimum of $\theta_{m,i}$ over all pairs $(m,i)$ in set $I$ and the angles between all pairs of $z_m, z_{m'}$, represented mathematically as:

$$\theta_0 := \min\{\min_{(m,i)\in I}\{\theta_{m,i}\}, \min_{m,m'\in M}\arccos(z_m \cdot z_{m'})\} \tag{10}$$

Upon normalization, any pair of distinct embeddings will maintain an angular separation of at least $\theta_0$. Then we have

$$
\begin{aligned}
\mathcal{L}_{\text{SupReMix}} &= \sum_{m\in M}\frac{1}{k_m}\sum_{i=1}^{k_m}\sum_{\substack{j=1\\j\neq i}}^{k_{(m,i),m}}\log\left(\frac{\sum_{\overline{m}\neq m}\sum_{l=1}^{\prime k_{(m,i),\overline{m}}}w_{m,\overline{m}}\exp(\langle z_{m,i}, z_{\overline{m},l}\rangle/\tau)}{\exp(1/\tau)}\right. \\
&\quad \left. + \frac{\sum_{\substack{l=1\\l\neq i}}^{k_{(m,i),m}}\frac{1}{m_{\max}-m_{\min}}\exp(1/\tau)}{\exp(1/\tau)}\right) \\
&< \sum_{m\in M}\frac{1}{k_m}\sum_{i=1}^{k_m}\sum_{\substack{j=1\\j\neq i}}^{k_{(m,i),m}}\log\left(\frac{\sum_{\overline{m}\neq m}\sum_{l=1}^{k_{(m,i),\overline{m}}}\exp(\cos(\theta_0)/\tau)+\frac{(k_{(m,i),m}-1)\exp(1/\tau)}{m_{\max}-m_{\min}}}{\exp(1/\tau)}\right) \\
&= \mathcal{L}^* + \sum_{m\in M}\frac{1}{k_m}\sum_{i=1}^{k_m}\sum_{\substack{j=1\\j\neq i}}^{k_{(m,i),m}}\log\left(1+\frac{(m_{\max}-m_{\min})\sum_{\overline{m}\neq m}\sum_{l=1}^{k_{(m,i),\overline{m}}}\exp((\cos(\theta_0)-1)/\tau)}{(k_{(m,i),m}-1)}\right)
\end{aligned}
\tag{11}
$$

Now since $\tau\to\infty$, we have the log functions in the second term converge to 0, then for any $\epsilon>0$ there exists $\tau_0>0$, for all $\tau>\tau_0$, we have

$$\sum_{m\in M}\frac{1}{k_m}\sum_{i=1}^{k_m}\sum_{\substack{j=1\\j\neq i}}^{k_{(m,i),m}}\log\left(1+\frac{(m_{\max}-m_{\min})\sum_{\overline{m}\neq m}\sum_{l=1}^{k_{(m,i),\overline{m}}}\exp((\cos(\theta_0)-1)/\tau)}{(k_{(m,i),m}-1)}\right)<\epsilon, \tag{12}$$

therefore, we have

$$\mathcal{L}_{\text{SupReMix}} < \mathcal{L}^* + \epsilon.$$

Based on the above and the proof of lemma 1, we know that the loss function is closed to its infimum if, and only if, the following are true:

1. all the real samples with the same label $m$ are embedded close to some vector $z_m$;

2. all Mix-pos of an anchor $(m,i)$ have embeddings close to $z_m$;

3. all the negatives (real and Mix-neg) of an anchor $(m,i)$ have $z_{m,i}\cdot z_{m',j}$ that are not equal to 1.

Conditions 1 and 2 are both necessary and sufficient for the inequality in equation 9 to closely approach equality. On the other hand, condition 3, specified as $\cos\theta_0\neq 1$, is the necessary and sufficient condition for inequality in equation 11 — an angular version of equation 9 — to similarly approach equality. Moreover, condition 2 is true when for any anchor $(m,i)$, all the samples with label $m'$ such that $m-\epsilon<m'<m+\epsilon$ are closed to a line, and their positions on the line are proportional to their labels. In other words, they are locally ordered and linear. Finally, condition 3 holds when the negative pairs are apart from each other, combined with condition 2 shows $z_m$ are globally ordered. Therefore, we can see the loss function approaches its infimum when the embeddings are globally ordered and locally linear. $\square$

# B DATASET

## B.1 DATASET DETAILS

Table 12: Overview of the six curated datasets used in our experiments

| Dataset | Target type | Target range | Bin size | # Training set | # Val. set | # Test set |
|---------|-------------|--------------|----------|----------------|------------|------------|
| IMDB-WIKI | Age | 0∼186* | 1 | 191,509 | 11,022 | 11,022 |
| AgeDB-DIR | Age | 0∼101 | 1 | 12,208 | 2,140 | 2,140 |
| STS-B-DIR | Text similarity score | 0∼5 | 0.1 | 5,249 | 1,000 | 1,000 |
| UK Biobank | Brain age | 42∼82 | 1 | 19,509 | 2,434 | 2,431 |
| HCP-Aging | Brain age | 36∼100 | 1 | 456 | 100 | 100 |
| UCI-airfoil | Scaled Sound Pressure Level | 103∼141 | 1 | 1,203 | 150 | 150 |

*Note: wrong labels in the original dataset.*

### B.1.1 IMDB-WIKI

The initial IMDB-WIKI dataset (Rothe et al., 2018), serves as a substantial repository of facial images intended for age determination through individual input images. This inaugural edition encompasses 523,000 facial photographs accompanied by the respective ages of the individuals pictured. A significant portion of this dataset (460,700 images) was amassed from the IMDB website, while the remaining 62,300 images were sourced from Wikipedia. After filtering out nqualified images with low face scores (Rothe et al., 2018; Yang et al., 2021), we randomly split the dataset into 191.5k images for training, 11k images for validation, and 11k images for testing. During the data pre-processing stage, we resize the images to a resolution of 224 x 224 pixels and normalize all the image values to fall within a range of 0 to 1.

### B.1.2 AGEDB-DIR

The original AgeDB dataset (Moschoglou et al., 2017) is a manually collected in-the-wild age database with accurate and clean labels. We follow the imbalanced split from Yang et al. (2021) to curate the dataset with 12,208 images for training, 2140 images for validation, and 2140 images for testing. Similar to IMDB-WIKI, the images in AgeDB are resized to 224 x 224 and normalized.

### B.1.3 STS-B-DIR

The Semantic Textual Similarity Benchmark (STS-B) (Cer et al., 2017), also included in the GLUE benchmark (Wang et al., 2018), is a collection of sentence pairs drawn from news headlines with human annotation of the similarity score from 0 to 5, video and image captions, and natural language inference data. We follow the imbalanced split from Yang et al. (2021) to curate STS-B-DIR with 5249 pairs for training, 1000 pairs for validation, and 1000 pairs for testing. We use NLTK toolkit (Loper & Bird, 2002) tokenization and 300D GloVe word embeddings (Pennington et al., 2014) for processing the data.

### B.1.4 UK BIOBANK

UK Biobank is a large-cohort of brain imaging data from predominantly healthy participants (Allen et al., 2012; Miller et al., 2016). In this study, we use the T1 volumetric data from 24,374 subjects (we exclude subjects with current/past stroke, cancer, long standing illness and poor health rating), of which 19,509 are used for training, 2,434 for validation and 2,431 for testing. We use data as preprocessed by UK Biobank using the original pipeline described in ALF (2018). To reduce memory consumption, all images are downsampled to 2mm-input-resolution and center cropped into 100 x 100 x 100.

### B.1.5 HCP-AGING

We use resting state fMRI data from HCP-Aging, which aims to study the aging process from middle age to older adulthood. To preprocess the fMRI data, we use the CBIG pipeline and Yeo-400-atlas (Schaefer et al., 2018). Consequently, our input data is in the shape of [400 (# parcells), n (# time frames)].

## B.2 ETHIC STATEMENTS

All datasets used in our experiments are publicly available and do not contain private information.

- Access to UK Biobank data is available upon application via the website (https://www.ukbiobank.ac.uk/). For UK Biobank, all private information about subjects is de-identified.
- HCP-Aging dataset can be downloaded from (https://www.humanconnectome.org/lifespan-studies). We use resting-state fMRI that does not contain any private information after the pre-processing.
- For AgeDB, IMDB-WIKI, STS-B, and UCI-airfoil, datasets are accrued without any engagement or interference involving human participants and are devoid of any confidential information.

## C EXPERIMENTAL SETTINGS

### C.1 BASELINE METHODS

**Supervised Contrastive Learning (SupCon).** We adapt SupCon (Khosla et al., 2020) to our regression tasks by treating samples (within one mini-batch) with the same target as positive pairs with different targets as negative pairs. For UCI-airfoil and HCP-Aging, where data augmentation is not applied, we stop the gradient update from samples that have no positive pair. We use a two-layer MLP as the projection head on all datasets following Chen et al. (2020a).

**Simple Contrastive Learning of Sentence Embeddings (SimCSE).** We adapt supervised SimCSE (Gao et al., 2021) to regression tasks by binarizing our continuous targets with a threshold of 2. Therefore, pairs with similarity scores greater than 2 will be considered positive pairs and vice versa.

**Supervised Contrastive Regression (SupCR).** We re-implement SupCR following Zha et al. (2022), as their code is not available. We follow their stated hyper-parameters, except for the UK Biobank dataset, where we change the batch size to 128 due to memory limitations.

### C.2 IMPLEMENTATION DETAILS

Table 13: Hyper-parameters used in SupReMix. CJ: color jittering; RC: random crop; LP: linear probe; FT: fintune

| Dataset | IMDB-WIKI | AgeDB-DIR | STS-B-DIR | UK Biobank | HCP-Aging | UCI-airfoil |
|---|---|---|---|---|---|---|
| Temperature ($\tau$) | 0.3 | 0.5 | 1.0 | 0.5 | 0.5 | 1.0 |
| Data Augmentation | CJ + RC | CJ + RC | N/A | N/A | N/A | N/A |
| Evaluation Protocol | LP | LP | LP | FT | FT | LP |
| Beta Distribution | Beta(2, 8) | Beta(2, 8) | Beta(2, 8) | Beta(2, 8) | Beta(2, 8) | Beta(2, 8) |
| Window Size ($\gamma$) | 5 | 5 | 1.0 | 3 | 5 | 7 |
| Backbone Network ($f(\cdot)$) | ResNet-50 | ResNet-50 | BiLSTM | 3D ResNet-18 | 1D ResNet-18 | 3layer MLP |
| Projection Head | 2layer MLP | 2layer MLP | 2layer MLP | 2layer MLP | 2layer MLP | 2layer MLP |
| Feature Dim | 128 | 128 | 128 | 128 | 128 | 128 |
| Learning Rate | 1e-3 | 1e-3 | 1e-3 | 1e-3 | 1e-3 | 1e-3 |
| Batch Size | 64 | 128 | 128 | 64 | 64 | 32 |

# D    ADDITIONAL RESULTS AND ANALYSIS

## D.1    FROM A PROXY TASK PERSPECTIVE

In Section 4, we quantitatively establish the superior performance of SupReMix compared to other supervised contrastive learning baselines. This superiority is largely attributed to its ability to induce a number of hard contrastive pairs incorporating the ordinal nature of regression. Figure 5 illustrates the superiority of SupReMix from the proxy task perspective (i.e. samples with the same label stay closer and samples with different labels stay apart). It shows a consistently widening gap between average positive and negative logits (based on real samples from AgeDB) over the training period — a gap notably larger than that achieved by SupCon. This suggests that SupReMix can produce representations that better align with our proxy task.

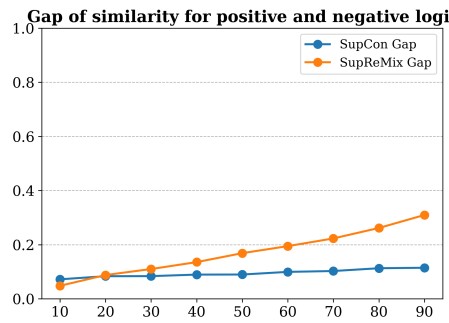

Figure 5: Gap of average positive and negative logits

## D.2    VARYING BATCH SIZE

In the realm of contrastive learning, pioneered by studies such as Chen et al. (2020a) and the initial implementation of supervised contrastive learning (Khosla et al., 2020), the deployment of large batch sizes has been a common strategy. This approach primarily facilitates the maintenance of an extensive pool of negative samples during loss computation. In our study, we delve into the impact of varying batch sizes in contrastive learning, particularly in the context of regression tasks. We use the AgeDB dataset to test the performance of SupCon and SupReMix with batch sizes ranging from 64 to 512. Our findings illustrate that the previously established notion — that increasing the batch size enhances performance — does not hold true in our case. This could potentially stem from the fundamental disparities between classification and regression methodologies.

Table 14: Effect of varying batch size for contrastive learning for regression

| Batch Size | MAE ↓ | | GM ↓ | |
|---|---|---|---|---|
| | SupCon | SupReMix | SupCon | SupReMix |
| 64 | 8.23 | 7.19 | 5.31 | 4.60 |
| 128 | 8.07 | 7.12 | 5.07 | 4.58 |
| 256 | 8.72 | 7.23 | 5.71 | 4.60 |
| 512 | 8.00 | 7.09 | 5.16 | 4.56 |

## D.3    GENERALIZATION ON MISSING TARGETS

Regression datasets often suffer from "missing targets", where samples with certain target values are absent in the training set. To explore the performance on this scenario for our proposed method, we curate the IMDB-WIKI, creating unseen targets by removing samples with ages in the ranges of 20-25, 50-55, and 75-80, while maintaining the original validation and test sets.

Table 15 illustrates that SupReMix significantly outperforms the Vanilla approach overall, improving MAE by 5.3%. Remarkably, it boosts performance under missing targets setting by 22.8%. This is due to SupReMix's ability to learn more continuous representations, leveraging our mixtures as effective landmarks for learning representations of missing targets, and enhancing prediction accuracy in scenarios with unseen data.

## D.4    "DISCRETIZATION" ALTERNATIVE FOR SUPERVISED CONTRASTIVE REGRESSION

One common strategy for tackling regression using classification techniques is discretization (Niu et al., 2016; Zhang et al., 2022). This involves converting continuous values into discrete "bins" to frame the issue as a classification problem. We examined this approach here, adapting this strategy to classification-derived contrastive learning to solve regression tasks. As depicted in Table 16, we varied the bin size from 1 (retaining the original label) to

Table 15: Evaluation on IMDB-WIKI with missing targets; MT: missing targets.

| | MAE ↓ | | GM ↓ | |
|---|---|---|---|---|
| | Overall | MT | Overall | MT |
| Vanilla | 5.81 | 7.54 | 3.49 | 5.99 |
| SupCon (Khosla et al., 2020) | 5.72 | 6.86 | 3.41 | 4.52 |
| SupCR (Zha et al., 2022) | 6.00 | 7.36 | 3.69 | 5.18 |
| SupReMix | 5.52 | 6.14 | 3.23 | 3.80 |
| GAINS (**Ours** VS. Vanilla(%)) | **+5.3** | **+22.8** | **+8.0** | **+57.6** |

20 during the contrastive learning stage of SupCon on AgeDB-DIR. The results demonstrate a decline in overall performance with the introduction of discretization; a decline that intensifies with larger bin sizes. This suggests that simply converting targets into a discretized format can negatively affect representation learning in regression tasks, hampering the natural continuity of the data.

Table 16: Results on varying bin size for supervised contrastive learning on AgeDB-DIR

| Bin Size | MAE ↓ | GM ↓ |
|---|---|---|
| 1 | 8.23 | 5.31 |
| 5 | 8.29 | 5.39 |
| 10 | 8.50 | 5.52 |
| 20 | 9.07 | 5.93 |

## D.5 COMPARISON WITH EXISTING REGRESSION LEARNING METHOD

We add a thorough performance comparison experiment with existing Mixup methods (Zhang et al., 2018; Verma et al., 2019; Guo et al., 2019; Barbano et al., 2023) and regression learning methods (Yao et al., 2022; Niu et al., 2016; Shin et al., 2022) Below we conducted two additional experiments using the AgeDB dataset: (1) we evaluated the regression performance of SupReMix (with pre-training followed by a linear probe) in comparison to Mixup methods and regression learning methods and achieved better performance than all of them; (from the first two parts of the table) (2) we demonstrated that integrating SupReMix with regression learning methods (with pretraining followed by linear probe with regression learning methods) can further enhance their performance (from the bottom part of the table below).

Table 17: Performance comparision of regression learning methods, prepresentation learning methods and their join combinations.

| Metrics | MAE↓ | MSE↓ | GM↓ |
|---|---|---|---|
| *Regression Learning (Train from starch)* | | | |
| Vanilla | 7.77 | 101.60 | 5.05 |
| Mixup (Zhang et al., 2018) | 8.03 | 119.51 | 5.78 |
| M-Mixup (Verma et al., 2019) | 8.02 | 104.33 | 5.34 |
| local-Mixup (Guo et al., 2019) | 7.89 | 102.33 | 5.19 |
| AdaMixup (Baena et al., 2022) | 7.92 | 103.21 | 5.23 |
| C-Mixup (Yao et al., 2022) | 7.66 | 99.82 | **4.82** |
| Ordinal (Niu et al., 2016) | 7.59 | 97.51 | 4.99 |
| MWR (Shin et al., 2022) | **7.47** | **93.26** | 4.83 |
| *Representation Learning (Linear Probing)* | | | |
| Mochi (Kalantidis et al., 2020) | 9.07 | 132.50 | 5.93 |
| i-Mix (Lee et al., 2020) | 8.50 | 117.96 | 5.52 |
| SupCon (Khosla et al., 2020) | 8.23 | 113.91 | 5.31 |
| SupCR (Zha et al., 2022) | 7.86 | 102.84 | 5.15 |
| **SupReMix** (ours) | **7.12** | **87.65** | **4.58** |
| *SupReMix + Regression Learning* | | | |
| SupReMix + Ordinal | 7.35 | 93.42 | 4.72 |
| SupReMix + MWR | **7.07** | **85.33** | **4.51** |
| SupReMix + C-Mixup | 7.17 | 90.21 | 4.79 |
| GAINS (**Best Joint** VS. Vanilla(%)) | **+9.0** | **+16.0** | **+10.7** |
| GAINS (**Best Joint** VS. Best RegL(%)) | **+5.2** | **+8.2** | **+6.4** |

## D.6 COMPARISION WITH C-MIXUP

We also performed additional performance comparisons between SupReMix with C-Mixup (Yao et al., 2022) across four different datasets. Here we show that SupReMix can consistently outperform C-Mixup train from scratch and bring improvement for C-Mixup with the SupReMix pretrain (bottom line of the table below).

Table 18: Performance comparison of SupReMix with C-Mixup on four datasets.

| Metrics (MAE↓) | AgeDB | IMDB | HCP | UKB |
|---|---|---|---|---|
| C-Mixup | 7.66 | 5.43 | 9.07 | 3.00 |
| **SupReMix** | **7.12** | 5.38 | 8.43 | 2.97 |
| **SupReMix** + C-Mixup | 7.17 | **5.21** | **8.23** | **2.90** |
| GAINS (**Joint** VS. C-Mixup(%)) | **+6.4** | **+4.1** | **+9.3** | **+3.3** |

## D.7 EXPERIMENT FOR GAZE ESTIMATION (MPIIFACEGAZE)

Below we show the performance comparison of SupReMix (w/o Mix-pos) with Vanilla and other contrastive learning counterparts (SupCon and SupCR). We carried our gaze estimation tasks with the subsampled (19500 for train, 1500 for validation, 1500 for test) MPIIFaceGaze dataset to evaluate the effectiveness of SupReMix.

We adhered to the model implementation as described by Zhang et al. (2017) and trained all methods from scratch. For SupCon and SupCR, we adopted the augmentation strategies outlined by Zha et al. (2022). Regarding the calculation of label distance, we utilized cosine distance, considering the multi-dimensional nature of our target. Our results demonstrate that SupReMix significantly outperforms the Vanilla baseline and other contrastive learning methods.

Table 19: Evaluation on MPIIFaceGaze

| Metrics | Angular ↓ | $R^2$ ↑ |
|---|---|---|
| Vanilla | 5.72 | 0.774 |
| SupCon (Khosla et al., 2020) | 6.01 | 0.767 |
| SupCR (Zha et al., 2022) | 5.48 | 0.801 |
| **SupReMix (w/o Mix-pos)** | **5.14** | **0.827** |
| GAINS (**Ours** VS. Vanilla(%)) | **+10.1** | **+6.8** |
| GAINS (**Ours** VS. SupCR(%)) | **+4.2** | **+3.2** |

## D.8 PAIR SELECTION STRATEGY

In our exploration of pair selection strategies, we focused on a comparative analysis involving our method versus the C-Mixup approach and input mixup. We conducted experiments on the AgeDB dataset. Specifically, we observed that the application of input mixup significantly diminishes the effectiveness of the SupReMix method. In contrast, our approach consistently demonstrated superior performance when compared to C-Mixup.

Table 20: Comparing different pair selection strategies

| Mixup Strategy | MAE ↓ | MSE ↓ | GM ↓ |
|---|---|---|---|
| Mixup (Zhang et al., 2018) | 10.18 | 154.64 | 6.74 |
| C-Mixup (Yao et al., 2022) | 7.22 | 92.68 | 4.83 |
| Ours | **7.12** | **87.65** | **4.58** |

## D.9 COMPARISION WITH GENERATIVE PRE-TRAINING BASELINES

We conduct a comparative analysis of our method against MAE He et al. (2022) and RoBERTa (Liu et al., 2019), focusing on linear-probe evaluations on AgeDB and STS-B datasets, respectively. In the case of MAE, we extend our comparison to both ViT-Based and ViT-Large models, each pre-trained on ImageNet. Our approach, SupReMix, demonstrates consistent superiority in performance over these methods.

Table 21: Comparison with generated pre-training method on AgeDB

| Mixup Strategy | MAE ↓ | MSE ↓ | GM ↓ |
|---|---|---|---|
| MAE (ViT-Base) (He et al., 2022) | 11.23 | 192.53 | 7.46 |
| MAE (ViT-large) (He et al., 2022) | 10.88 | 183.25 | 7.29 |
| SupReMix | **7.12** | **87.65** | **4.58** |

Table 22: Comparison with generated pre-training method on STS-B

| Metrics | $\rho$ ↑ | MAE ↓ | MSE ↓ |
|---|---|---|---|
| RoBERTa (Liu et al., 2019) | 0.751 | 0.777 | 0.987 |
| SupReMix | **0.760** | **0.759** | **0.893** |

## D.10 COMPARISON WITH OTHER CONTRASTING LEARNING (FINETUNING EVALUATION)

When comparing contrastive learning methods under a fine-tuning evaluation scheme with the entire backbone network fine-tuned.

Table 23: Fine-tuning evaluation on AgeDB

| Metrics | MAE ↓ | MSE ↓ | GM ↓ |
|---|---|---|---|
| SupCon (Khosla et al., 2020) | 8.83 | 123.36 | 5.77 |
| SupCR (Zha et al., 2022) | 8.12 | 110.31 | 5.21 |
| SupReMix | **7.23** | **93.66** | **4.69** |

Table 24: Fine-tuning evaluation on IMDB

| Metrics | MAE ↓ | MSE ↓ | GM ↓ |
|---|---|---|---|
| SupCon (Khosla et al., 2020) | 6.46 | 78.32 | 3,88 |
| SupCR (Zha et al., 2022) | 6.07 | 71.98 | 3.70 |
| SupReMix | **5.39** | **62.45** | **3.17** |

## E LIMITATION

The application of Mix-pos in this study faces limitations in its adaptability to higher-dimensional regression labels. This challenge stems from a fundamental issue: while ordinality is crucial in regression tasks, it is undefined for vectors in dimensions greater than one, as the topology on $\mathbb{R}^n$ does not form an order topology. In Mix-pos, obtaining a weights vector (whose dimension matches that of the label) by solving a linear system is theoretically feasible. However, this solution is not always assured due to potential linear independence between selected samples and the anchor. Moreover, the approach's scalability is hindered as the dimensionality of the regression label increases, posing significant practical challenges. Future research could focus on exploring methods to preserve ordinality in regression representations when dealing with higher-dimensional labels.

