# OpenReview forum: "Mixup Your Own Pairs"
_ICLR.cc/2024/Conference — Submitted to ICLR 2024_

### Official Review · Reviewer_ggUN · 2023-10-25

**Soundness:** 2 fair
**Presentation:** 4 excellent
**Contribution:** 2 fair
**Rating:** 3
**Confidence:** 5

**Summary:**

The paper discusses the neglect of contrastive learning for regression tasks. To tackle this issue, the paper introduces "Supervised Contrastive Learning for Regression with Mixup" (SupReMix). In particular, anchor-inclusive mixtures and anchor-exclusive mixtures are proposed. Evaluations are conducted on regression datasets, including 2D images, volumetric images, text, tabular data, and
time-series signals.

**Strengths:**

This paper is clearly written and organized. The idea of applying contrastive learning to regression is interesting.

**Weaknesses:**

* a) No sufficient comparisons are made to demonstrate the solidness of this work. Since this paper proposed a data augmentation method based on mixup, the performance of SupReMix should be compared with other mixup approaches and not just the baseline SupCon. Besides regular Mixup and manifold Mixup, there are many, many other mixup approaches, for instance, C-Mixup [1], local-Mixup [2], and manifold intrusion [3].

* b) Since the idea of contrastive learning is towards the clustering effect. One would naturally think that applying contrastive learning to regression is not viable. Then it would only make sense to compare SuperReMix to regression frameworks without contrative learning, for example, Moving Window Regression [4].

* c) Results in Tab. 1-6 show incremental improvement over the baseline. However, the gain is provided but is w.r.t to vanilla, not second best.

# References:
 [1] Yao, Huaxiu, et al. "C-mixup: Improving generalization in regression." Advances in Neural Information Processing Systems 35 (2022): 3361-3376.

[2] Baena, Raphael, Lucas Drumetz, and Vincent Gripon. "Preventing manifold intrusion with locality: Local mixup." arXiv preprint arXiv:2201.04368 (2022).

[3] Guo, Hongyu, Yongyi Mao, and Richong Zhang. "Mixup as locally linear out-of-manifold regularization." Proceedings of the AAAI conference on artificial intelligence. Vol. 33. No. 01. 2019.

[4] Shin, Nyeong-Ho, Seon-Ho Lee, and Chang-Su Kim. "Moving window regression: A novel approach to ordinal regression." Proceedings of the IEEE/CVF conference on computer vision and pattern recognition. 2022.

**Questions:**

See weakness.

---

> ### Author Response · Authors · 2023-11-16
> **Response to Reviewer ggUN**
>
> We would like to thank the reviewer for the valuable and constructive comments and suggestions. We have now performed new experiments and carefully revised our manuscript to address your concerns. We believe that the quality and clarity of our manuscript has been improved.
>
> $\textbf{a) and  b) More comparisons with Mixup methods and regression learning methods.}$ We thank the reviewer for pointing out the lack of comparison with other Mixup methods and regression learning methods. We firstly would like to hightlight that our SupReMix is a novel approach for $\textbf{representation learning}$ for regression tasks that stands apart from traditional $\textbf{regression learning}$ methods like C-Mixup[1]. As recommended, below we conducted two additional experiments using the AgeDB dataset:
> (1) we evaluated the regression performance of SupReMix (with pre-training followed by a linear probe) in comparison to Mixup methods and regression learning methods (with train from starch) including local-mixup[2], AdaMixup[3], C-Mixup[1], ordinal regression[5] and moving window regression (MWR)[4]. SupReMix has achieved better performance than $\textbf{ALL}$ of them; (from the first two parts of the table)
> (2) we demonstrated that integrating SupReMix with regression learning methods (with pretraining followed by linear probe with regression learning methods) can further enhance their performance (from the bottom part of the table below).
>
>
>  | Metrics                      | MAE ↓    | MSE ↓    | GM ↓    |
> | -----------------------------| -------- | -------- | ------- |
> | **Regression Learning (Train from scratch)** |
> | Vanilla                      | 7.77     | 101.60   | 5.05    |
> | local-Mixup                  | 7.89     | 102.33   | 5.19    |
> | AdaMixup                     | 7.92     | 103.21   | 5.23    |
> | C-Mixup                      | 7.66     | 99.82    | **4.82**|
> | Ordinal                      | 7.59     | 97.51    | 4.99    |
> | MWR                          | **7.47** | **93.26**| 4.83    |
> | **Representation Learning (Linear Probing)** |
> | SupCon                       | 8.23     | 113.91   | 5.31    |
> | SupCR                        | 7.86     | 102.84   | 5.15    |
> | **SupReMix (ours)**          | **7.12** | **87.65**| **4.58**|
> | **SupReMix + Regression Learning** |
> | SupReMix + Ordinal           | 7.35     | 93.42    | 4.72    |
> | **SupReMix + MWR**           | **7.07** | **85.33**| **4.51**|
> | SupReMix + C-Mixup           | 7.17     | 90.21    | 4.79    |
> | GAINS (**Best Joint** vs. Vanilla (%)) | **+9.0** | **+16.0**| **+10.7**|
> | GAINS (**Best Joint** vs. Best RegL (%)) | **+5.2** | **+8.2** | **+6.4** |
>
> We also performed additional performance comparisons between SupReMix with C-Mixup[1] across four different datasets (see the table below). Here we show that SupReMix can **consistently outperform C-Mixup** train from scratch and **bring improvement for C-Mixup** with the SupReMix pretrain (bottom row of the table below).
>
> | Metrics (MAE ↓)     | AgeDB  | IMDB  | HCP   | UKB   |
> | -------------------- | ------ | ----- | ----- | ----- |
> | C-Mixup              | 7.66   | 5.43  | 9.07  | 3.00  |
> | **SupReMix**         | 7.12   | 5.38  | 8.43  | 2.97  |
> | **SupReMix + C-Mixup** | **7.17** | **5.21** | **8.23** | **2.90** |
> | GAINS (Joint VS. C-Mixup(%)) | **+6.4** | **+4.1** | **+9.3** | **+3.3** |
>
> **c) Gain w.r.t to the Second Best.** Thank you for highlighting the importance of comparing our results with the second-best method, not just the vanilla baseline. Please note that vanilla is the second-best for some datasets (AgeDB, IMDB, HCP). For clarity, we have added underscores of the second-best performance metrics in Table 1-6.
>
> [5] Niu, Zhenxing, et al. "Ordinal regression with multiple output cnn for age estimation." Proceedings of the IEEE conference on computer vision and pattern recognition. 2016.

---

> > ### Author Response · Authors · 2023-11-19
> > **Request for Feedback on Our Response**
> >
> > Dear Reviewer,
> >
> > we would greatly appreciate your feedback on whether our response adequately addressed your questions and concerns.

---

> > > ### Author Response · Authors · 2023-11-22
> > > **Request for Engagement in Discussion**
> > >
> > > Dear Reviewer,
> > >
> > > As the discussion period ends tomorrow, we kindly request to know if our responses have addressed all your concerns. If so, could you please consider increasing your score? If there are further issues, we'd appreciate your further feedback.

---

> ### Author Response · Authors · 2023-11-22
> **discussion window close in less than one day, looking forward to your reply**
>
> Dear Reviewer ggUN,
>
> As the discussion period ends in **less than one day**, we kindly request to know if our responses have addressed all your concerns. If so, could you please consider increasing your score? We are more than happy to provide further clarification if time allows.

---

### Official Review · Reviewer_DdrH · 2023-10-27

**Soundness:** 3 good
**Presentation:** 2 fair
**Contribution:** 2 fair
**Rating:** 6
**Confidence:** 4

**Summary:**

The authors study how contrastive representation learning can be improved, specifically for regression problems.

They propose Supervised Contrastive Learning for Regression with Mixup (SupReMix). This entails utilizing mixup in the feature embedding space, to create harder positive and negative contrastive pairs. Hard negatives are created by mixing the anchor with a normal negative example. Hard positives are created by mixing two normal negative examples, such that if the corresponding regression targets are mixed in the same manner, this equals the target of the anchor.

The proposed method is applied to six datasets with different input modalities, with age, text similarity score or sound pressure level as the regression targets. Their method consistently outperforms the vanilla regression baseline, and mostly also other baselines for supervised contrastive learning.

**Strengths:**

I definitely agree with the authors that representation learning for _regression problems_ has received less attention than for classification. This is an important and interesting problem that I think should be studied more.

The overall proposed method makes intuitive sense, it seems like a reasonable approach.

The proposed method is shown to consistently improve the vanilla regression baseline across a quite wide range of applications.

**Weaknesses:**

The paper could be a bit more well-written overall. For example:
- Section 2 seems somewhat out-of-place. To me, it doesn't quite follow naturally from Section 1, and it is not entirely clear what the main takeaway should be or how it connects to the rest of the paper.
- Equation (1) and (3) are difficult to read/parse. They are quite "dense".
- It is not clear what the theoretical results in Section 3.4 are supposed to tell me, what should be my main takeaway?

The proposed method is only applied to regression problems with 1D targets. It is not clear to me whether or not it could be extended.

The proposed method is only applied to datasets where the regression targets take on a relatively small set of different values (see Table 12), for example age in the interval [0, 100] with a bin size of 1. It is not entirely clear to me whether or not the method is somehow limited to such regression problems.

**Questions:**

1. Could the proposed method be extended also to regression problems where the targets are multi-dimensional?

2. Is the proposed method somehow limited to regression problems such as age estimation, in which the targets take on a relatively small discrete set of values? Could it be applied also to problems with "truly continuous" regression targets? What happens if, within a given batch, it is not possible to find examples with exactly the same regression target as the anchor?


Minor things:
- Section 4, Evaluation Schemes: The linear probing protocol is used for 4 out of 6 datasets. Any particular reason why? Why not for all 6? Or, why do you finetune the whole network for those 2 particular datasets?
- Table 9 and 10 are interesting results, but for which dataset(s) are these results?
- Section 1, "its dependency on data augmentation restricts its applicability to domains where effective augmentation techniques are lacking": I think this should be the other way around, the applicability is restricted to domains where effective augmentation methods do exist?
- Figure 1 caption, "encodes input to embeddings": input --> inputs?
- Section 3, "which encodes input to embeddings": input --> inputs?
- Personally, I would probably consider modifying the title, making it a bit more descriptive (the current title gives no clear clue that the paper studies the important problem of representation learning for regression).

---

> ### Author Response · Authors · 2023-11-16
> **Response to Reviewer DdrH [1/2]**
>
> Thank you for recognizing the importance of the issue we tackle, appreciating our methodology, and acknowledging the consistent improvement we have achieved. We are very grateful for your constructive comments. Below, we carefully address each of your questions to clarify and improve our submission.
>
> **Improve the clarity of our paper:**
> As recommended, we have carefully revised the text (as well as figures and equations) to improve clarity and readability as detailed below.
>
> - For Section 2, the main motivation is to show that the direct application of contrastive learning for classification to regression can only learn **discontinuous representations that are insensitive to label permutation** (Section 2.1) and **swift saturation** of training with conventional positive/negative contrastive pairs (Section 2.2). ("various interesting empirical studies of regression tasks, e.g., visualizations of logit distribution and latent space. These findings are supportive of the proposed method and might be inspiring for designing better algorithms for regression tasks." from reviewer 4GQV). Section 2.1 illustrates that SupCon is deficient in ordinality-awareness as it consistently generates embeddings that exhibit discrete clustering patterns, regardless of whether the labels are genuine or permuted. In Section 2.2, we observe that SupCon also lacks in presenting sufficient hardness; notably fewer contrastive pairs contribute to training during the middle or late stages. These observations directly relate to the two key aspects we previously highlighted: ordinality-awareness and hardness. To address these issues, we introduce Mix-pos and Mix-neg with weights. This enhancement aims to equip the model with a better grasp of ordinality and to amplify the level of hardness in contrastive pairs. We have modified the text to make it more explicit.
> - For equation (1) and equation (3), we have made some modifications to **parse them** further for better readability. We change the previous set of expressions into:
>
> $z_{m,i}^{-} = \lambda_1 \cdot z_{m,i} + (1-\lambda_1) \cdot z', \quad
> \text{where} \ z' \in I\backslash I_m, \ \lambda_1 \sim \text{Beta}(\alpha, \beta), \
> \text{and} \ \overline{m} = \lambda_1 \cdot m + (1-\lambda_1) \cdot m'$ and
> $z_{m,i}^{+} = \lambda_2 \cdot z_{m',i'} + (1-\lambda_2) \cdot z_{m'',i''}, \quad
> \text{where} \ i'<i<i'', \ i-i', i''-i \leq \gamma, \
> \text{and} \ \lambda_2 \cdot m' + (1-\lambda_2) \cdot m'' = m$
>
>
> - The theoretical results in Section 3.4 are "theoretical explanations" (reviewer 4GQV) for our SupReMix loss function. In Theorem 1, the key insight is that SupReMix amplifies the penalty for more distant negative pairs compared to closer ones by assigning specific weights to these pairs. Meanwhile, Lemma 1 and Theorem 2 collectively highlight that the SupReMix loss function not only possesses a tight bound but also ensures both the ordinality and continuity of representations as they converge towards the infimum. We have concisely summarized the main takeaway in Section 3.4 to more effectively communicate the key message of this section.

---

> ### Author Response · Authors · 2023-11-16
> **Response to Reviewer DdrH [2/2]**
>
> **Higher Dimensional Output:**
> Thank you for this important feedback. Firstly, even if ordinality has been shown to be important in regression tasks, it is undefined for vectors in dimensions greater than one. The topology on $\mathbb{R}^n$ does not constitute an order topology, making the application of our Mix-pos method to outputs in dimensions greater than one non-trivial. In Mix-pos, obtaining a weights vector (whose dimension matches that of the label) by solving a linear system is theoretically feasible. However, this solution is not always assured due to the potential linear independence between selected samples and the anchor. And it is not scalable when increasing the dimension of regression labels. Future research could focus on exploring methods to preserve ordinality in regression representations when dealing with higher-dimensional labels. However, our Mix-neg approach and weights design can be straightforwardly applied to higher-dimensional settings. To illustrate this, we have included additional experiments to perform regression in gaze prediction (two outputs), demonstrating that our Mix-neg and weight adjustments can lead to significant improvements. We have added one more limitation section (Section E) in our submission to discuss this point.
>
> Below we show the performance comparison of SupReMix (w/o Mix-pos) with Vanilla and other contrastive learning counterparts (SupCon and SupCR). We carried out gaze estimation tasks with the subsampled (due to the time limitation) MPIIFaceGaze dataset to evaluate the effectiveness of SupReMix. In terms of the calculation of label distance, we use cosine distance since the dimension of the target is larger than one. (More details about the experiment are in Appendix D.7.) We show that SupReMix significantly outperforms Vanilla and other contrastive learning methods.
>
> | Metrics               | Angular ${\downarrow}$ | $R^{2}$ ${\uparrow}$ |
> | ----------------------| ----------------------- | --------------------- |
> | Vanilla               | 5.72                    | 0.774                 |
> | SupCon                | 6.01                    | 0.767                 |
> | SupCR                 | 5.48               | 0.801             |
> | **SupReMix (w/o Mix-pos)** | **5.14**      | **0.827**             |
> | GAINS (**Ours** VS. Vanilla(%)) | **+10.1** | **+6.8** |
> | GAINS (**Ours** VS. SupCR(%))   | **+4.2**  | **+3.2** |
>
> **Truly Continuous:**
> "Could it be applied also to problems with 'truly continuous' regression targets?" The answer is a definite **YES**. SupReMix could be adapted to **ANY** bin size within the target range. In fact, smaller bin sizes are particularly advantageous as they promote greater continuity in the representations. This is due to the increased diversity of candidates available for the mixup process. Crucially, SupReMix does not depend on having samples with identical labels. It functions effectively even in scenarios where continuous variation is extreme, such as cases where each sample has a unique label. For Mix-neg, the anchor is combined with all other samples that bear different labels. In the case of Mix-pos, samples within a specified window and possessing distinct labels from the anchor are chosen for mixup. An important feature of this approach is its ability to always find a positive weight that combines a larger and a smaller label to match the anchor's label. This is achieved by simply solving a linear equation with one variable. The design of SupReMix thus ensures both flexibility and precision in handling a wide range of label variations. The bin sizes selected for our experiments were determined by the inherent characteristics of each dataset, independent of the design of our model.
>
> **Minor Things:**
>
> - For the neuroimaging datasets (HCP and UKB), we followed the previous work [1] and [2] that fine-tuned the whole network.
>
> - Table 9 was based on AgeDB. We have added the information in the tables. Table 10 was based on AgeDB and STS-B (indicated in the header row).
>
> - As suggested, we have changed the sentence to "Nevertheless, its dependency on data augmentation restricts its applicability to domains where effective augmentation techniques have been established, such as time series or tabular data."
>
> - We have changed the "input" to "inputs," thank you for pointing this out.
>
> - We have changed our title to "Mixup Your Own Pairs: Supervised Contrastive Learning for Regression with Mixup." Thank you for your excellent suggestion.
>
> [1] Hongming Li, Theodore D Satterthwaite, and Yong Fan. Brain age prediction based on resting-state functional connectivity patterns using convolutional neural networks. ISBI 2018, pp. 101–104.
> [2] Benedikt Atli J ́onsson, Gyda Bjornsdottir, TE Thorgeirsson, Lotta Mar ́ıa Ellingsen, G Bragi Walters, DF Gudbjartsson, Hreinn Stefansson, Kari Stefansson, and MO Ulfarsson. Brain age prediction using deep learning uncovers associated sequence variants. Nature communications,  2019.

---

> > ### Author Response · Authors · 2023-11-19
> > **Request for Feedback on Our Response**
> >
> > Dear Reviewer,
> >
> > we would greatly appreciate your feedback on whether our response adequately addressed your questions and concerns.

---

> > > ### Author Response · Authors · 2023-11-22
> > > **Request for Engagement in Discussion**
> > >
> > > Dear Reviewer,
> > >
> > > As the discussion period ends tomorrow, we kindly request to know if our responses have addressed all your concerns. If so, could you please consider increasing your score? If there are further issues, we'd appreciate your further feedback.

---

> > > > ### Comment · Reviewer_DdrH · 2023-11-22
> > > > **Response to rebuttal**
> > > >
> > > > (Sorry for my late response. I have struggled to find enough time to both write responses as an author, and participate in the discussions as a reviewer)
> > > >
> > > > Thank you for your detailed response.
> > > >
> > > > I have read the other reviews and all responses. My concerns have been addressed quite well.
> > > >
> > > > I have increased my score from "5: marginally below" to "6: marginally above".

---

### Official Review · Reviewer_gvot · 2023-11-04

**Soundness:** 3 good
**Presentation:** 2 fair
**Contribution:** 3 good
**Rating:** 8
**Confidence:** 3

**Summary:**

The paper studies representation learning in the context of regression, which is a problem that has received relatively lesser attention in literature. Motivated by the fact that typical objectives for representation learning often introduce unwanted inductive biases into the learning, the paper argues for two key objectives to consider when setting up the represnetation learning : (a) ordinality-awareness, i.e., the latent space must reflect or be sensitive to the ordinality of the output and (b) hardness, i.e., choosing the right level of hardness in choosing negatives and positives within a contrastive training setup.

Ordinality is enforced by choosing negatives from an ordered set of samples according to the function value, such that samples in a level set (same value) are mixed up with samples with different labels.

Hard positives are chosen by mixing up samples that have function value above and below the anchor's function value, where the mixup coefficient is chosen such that the linear combination in the output equals the anchor's actual function value.

Experiments on a number of different regression benchmarks show improvements over related baselines.

**Strengths:**

* The paper addresses an important problem on improving representation learning for regression, a domain to which a lot of our insights and beliefs from classification may not generalize.
* I think formulation is intuitive and clever, exploiting the ordinality and mixup in choosing hard positives and negatives makes sense and appears to help.
* Importantly, the proposed approach -- SupReMix leverages these without leveraging augmenting functions, which is standard in most representation learning to obtain different views of a sample. This is significant since its non-trivial to choose augmentations in regression or for time series data.
* SupReMix shows superior performance across all baselines considered on all or most of the benchmarks which is encouraging.

**Weaknesses:**

* I think the biggest weakness is the lack of comparisons with related works in regression -- most notably C-Mixup (Yao et al., NeurIPS 2022) -- and vanilla mixup as well, as it has been shown to be a competitive baseline even on regression. I think they may be better than the ERM (vanilla) baseline in the experiments. Besides, the C-mixup method is also closely related in the sense that they choose mixup pairs that are close in function value (albeit without considering ordinality) so its worth a deeper comparison even methodologically. It might provide some insights into the strengths of SupReMix.  For example, does adding ordinality with vanilla mixup or C-mixup boost performance of each method?
* The framework fundamentally relies on ordering the training data according to function value -- how is this implemented in practice? Is the ordering done on each mini batch?
* Are the backbone networks trained from scratch or do you use pre-trained variants followed by fine-tuning?
* A limitation that is unaddressed here, is that the current method does not generalize to even simpler regression problems where the output function has dimensionality $>1$.
* Finally, I find the notational style in the paper is a bit confusing and not easy to read. It may help the reader to simplify or modify the notation to benefit legibility .. for e.g. indices, function values, embeddings are all in italics which is not typical and hard to distinguish. Please fix these.
* Figures 1 A and B, do not convey the message of the method clearly.. and is a bit confusing. I think this can be improved to make the proposed method more explicit.

**Questions:**

please see questions above.

---

> ### Author Response · Authors · 2023-11-16
> **Response to Reviewer gvot [1/2]**
>
> Thank you for recognizing the significance of the problem we address, the ingenuity of our formulation, and the importance of our method, which notably does not depend on data augmentation and demonstrates superior performance. We truly appreciate your helpful suggestions and have provided detailed responses/revisions to address your points.
>
> **More Comparisons (C-Mixup):**
> Thank you for your constructive comment. As recommended, we have added performance comparison with C-Mixup (training-from-scratch) on four datasets. Moreover, as SupReMix is a representation learning method, it can **jointly** work with C-Mixup by using SupReMix pretraining and C-Mixup during the linear probe. We demonstrate that, compared to training from scratch, utilizing our SupReMix pre-training consistently yields improvements.
>
> | Metrics (MAE↓) | AgeDB | IMDB | HCP | UKB |
> | -------------- | ----- | ---- | --- | --- |
> | C-Mixup | 7.66 | 5.43 | 9.07 | 3.00 |
> | **SupReMix** | **7.12** | 5.38 | 8.43 | 2.97 |
> | **SupReMix + C-Mixup** | 7.17 | **5.21** | **8.23** | **2.90** |
> | GAINS (Joint VS. C-Mixup(%)) | **+6.4%** | **+4.1%** | **+9.3%** | **+3.3%** |
>
> **Pair Selection Strategy:**
> We appreciate your in-depth advice on the relation of SupReMix with C-Mixup in terms of "pair selection strategy". In short, our pair selection strategy is in the same direction but different from C-Mixup's. Although both of us consider the label distance, C-Mixup will only put a higher probability of mixup for pairs with smaller label distances. SupReMix, as a contrastive learning method, treats positive pairs mixup (Mix-pos) and negative pairs mixup (Mix-neg) differently. More specifically, SupReMix takes a window size (γ) of label distance for Mix-pos and leaning-toward-anchor beta distribution for sampling anchor-inclusive pairs (Mix-neg). Mix-Pos strongly depends on local linearity assumption, therefore we do not consider samples with large label distance (even with low probability) for hard positive mixtures. For performance comparison, we replaced our pair selection strategy with C-Mixup's and experimented on the AgeDB dataset. We have shown that our proposed approach still outperforms C-Mixup (see the table below).
>
> | Mixup Strategy | MAE↓ | MSE↓ | GM↓ |
> | -------------- | ---- | ---- | --- |
> | C-Mixup | 7.22 | 92.68 | 4.83 |
> | Ours | **7.12** | **87.65** | **4.58** |
>
> **No Ordering is Needed:**
> Thanks for raising this question. We would like to clarify that our method does not require a specific order for selecting contrastive pairs. For hard negatives, the anchor is mixed with all other samples in the same mini-batch that have differing labels. Regarding hard positives, the selection process involves identifying samples within a predefined γ window. This is achieved by first calculating the label distance between the anchor and all other samples. We then select the samples that fall within the window. This process can be efficiently implemented through matrix subtraction and identifying the minimum values.
>
> **All the backbone networks were trained from scratch (if you refer to the contrastive learning stage itself).**

---

> ### Author Response · Authors · 2023-11-16
> **Response to Reviewer gvot [2/2]**
>
> **Higher Dimensional Output:**
> Thank you for this important feedback. Firstly, even if ordinality has been shown to be important in regression tasks, it is undefined for vectors in dimensions greater than one. The topology on $\mathbb{R}^n$ does not constitute an order topology, making the application of our Mix-pos method to outputs in dimensions greater than one non-trivial. In Mix-pos, obtaining a weights vector (whose dimension matches that of the label) by solving a linear system is theoretically feasible. However, this solution is not always assured due to the potential linear independence between selected samples and the anchor. And it is not scalable when increasing the dimension of regression labels. Future research could focus on exploring methods to preserve ordinality in regression representations when dealing with higher-dimensional labels. However, our Mix-neg approach and weights design can be straightforwardly applied to higher-dimensional settings. To illustrate this, we have included additional experiments to perform regression in gaze prediction (two outputs), demonstrating that our Mix-neg and weight adjustments can lead to significant improvements. We have added one more limitation section (Section E) in our submission to discuss this point.
>
> Below we show the performance comparison of SupReMix (w/o Mix-pos) with Vanilla and other contrastive learning counterparts (SupCon and SupCR). We carried our gaze estimation tasks with the subsampled (due to the time limitation) MPIIFaceGaze dataset to evaluate the effectiveness of SupReMix. In terms of the calculation of label distance, we used cosine distance since the dimension of the target is larger than one. (More details about the experiment are in Appendix D.7.) We show that SupReMix **significantly outperforms** Vanilla and other contrastive learning methods.
>
> | Metrics | Angular↓ | R²↑ |
> | ------- | -------- | --- |
> | Vanilla | 5.72 | 0.774 |
> | SupCon | 6.01 | 0.767 |
> | SupCR | 5.48 | 0.801 |
> | **SupReMix (w/o Mix-pos)** | **5.14** | **0.827** |
> | GAINS (Ours VS. Vanilla(%))| **+10.1** | **+6.8** |
> | GAINS (Ours VS. SupCR(%))| **+4.2** | **+3.2** |
>
> **Notations:**
> Thank you for your advice. We have made the following modifications to enhance clarity and conformity:
> (1) The text in equations has been altered from italic to normal font. For instance, the word "with" in Theorem 1 now appears in standard font. (2) Vectors are now in bold, aligning with common mathematical practice, e.g., vector **z** used for embeddings. (3) The function notation g has been replaced with a Greek letter ρ.
>
> **Figures:**
> Thank you for your suggestion to improve the figure of schematic overview. We have made modifications to make the proposed method more explicit: (1) In panel A, we use red and green to color code the Mix-Pos and Mix-Neg; (2) In panel B, we change the names of our hard pairs to "Mix-Pos" and "Mix-Neg", to be consistent with panel A.

---

> > ### Author Response · Authors · 2023-11-19
> > **Request for Feedback on Our Response**
> >
> > Dear Reviewer,
> >
> > we would greatly appreciate your feedback on whether our response adequately addressed your questions and concerns.

---

> > > ### Comment · Reviewer_gvot · 2023-11-21
> > >
> > > Thanks for your responses, you have addressed all my concerns. I will vote to accept this paper.

---

### Official Review · Reviewer_4GQV · 2023-11-12

**Soundness:** 3 good
**Presentation:** 3 good
**Contribution:** 2 fair
**Rating:** 5
**Confidence:** 5

**Summary:**

This paper argues that the neglect of ordinality-awareness and hardness causes the sub-optimal performances of contrastive learning for regression. Specifically, the authors propose hard sample mining with mixup augmentations for contrastive learning in regression tasks, dubbed SupReMix, which takes anchor-inclusive mixtures as hard negative pairs and anchor-exclusive mixtures as hard positive pairs. Empirical and theoretic analyses are provided to demonstrate the effectiveness. Extensive experiments on various modalities show the performance gains of the proposed SupReMix over the Vanilla baseline and existing contrastive-based regression methods.

**Strengths:**

(**S1**) The proposed method is well-motivated to tackle the hard sample mining problem for regression tasks and achieves significant performance gains upon Vanilla.
(**S2**) The authors provide various interesting empirical studies of regression tasks, e.g., visualizations of logit distribution and latent space. These findings are supportive of the proposed method and might be inspiring for designing better algorithms for regression tasks.
(**S3**) The overall representations of the manuscript are well-ranged and easy to follow, which provides empirical analysis and theoretic explanations.

**Weaknesses:**

(**W1**) Lack of novelty compared to existing methods. The studied problem and proposed method are not novel enough. As known to all, the hard sample mining problem has been explored since 2020, there are also some relevant methods for regression methods (SupCon, SupCR, Decoupled CL [1]). Meanwhile, the proposed SupReMix utilizes similar strategies to design hard samples as MoCHi [2] and i-Mix [3] for contrastive pre-training and uses similar anchor and sample selection strategies as C-Mixup [4].

(**W2**) Weak experiments. Despite the authors providing extensive comparison results with two directly relevant contrastive learning methods, more related baselines should be compared in various experimental settings. Firstly, the authors adopt the implementation from C-Mixup [4] while not comparing it with it. I suggest the authors add more comparison experiments with existing mixup methods for regression (e.g., Vanilla Mixup, C-Mixup [4], ManifoldMix [5], etc) and general hard sample mining contrastive pre-training methods with mixup augmentations (e.g., MoCHi [2] and i-Mix [3]). Secondly, the authors should not only evaluate on self-supervised pre-training and fine-tuning setting but should conduct the practical training-from-scratch setting as well.

(**W3**) Usage of hyper-parameters. Firstly, many hyper-parameters are used in SupReMix compared to SupCon and mixups for regression tasks. The most widely used mixup augmentations are

(**W4**) Overlook of some related works. As mentioned above, some general and relevant mixup augmentations [3, 5, 6, 7] should be included as background knowledge. Meanwhile, relevant hard sample mining techniques [2, 5, 7] in contrastive learning should be included and discussed in the related work section.

### Reference
[1] Decoupled Contrastive Learning. In ECCV, 2022.

[2] Hard Negative Mixing for Contrastive Learning. In NeurIPS, 2020.

[3] i-Mix: A Domain-Agnostic Strategy for Contrastive Representation Learning. In ICLR, 2021.

[4] C-Mixup: Improving Generalization in Regression. In NeurIPS, 2022.

[5] Harnessing Hard Mixed Samples with Decoupled Regularizer. In NeurIPS, 2023.

[6] Manifold Mixup: Better Representations by Interpolating Hidden States. In ICML, 2019.

[7] Un-Mix: Rethinking Image Mixtures for Unsupervised Visual Representation Learning. In AAAI, 2022.

**Questions:**

Please refer to the weaknesses.

---

> ### Author Response · Authors · 2023-11-16
> **Response to Reviewer 4GQV [1/2]**
>
> Thank you for recognizing the strong motivation behind our method and its significant performance improvements. We also appreciate your acknowledgment of our comprehensive empirical analysis and theoretical explanations. We have prepared the response below following your constructive comments and suggestions. We hope that the following has addressed your concerns adequately.
>
> **Novelty (W1):**
>
> We appreciate the reviewer's assessment regarding the novelty of our proposed method. It provides us with an opportunity to clarify the contributions and innovative aspects of our work. The novelty of our work can be summarized in the following three key aspects:
>
> **(1) Contribution to an Underexplored Domain:**
> We wish to highlight that the majority of previously proposed hard sample mining methods, including those like SupCon and Decoupled CL you mentioned, predominantly focus on classification rather than **regression**. The direct application of them to regression leads to sub-optimal performance due to lack of ordinality-awareness and hardness. Regression is "a domain to which a lot of our insights and beliefs from classification may not generalize" (reviewer gvot) and "an important and interesting problem that I think should be studied more" (reviewer DdrH). "The idea of applying contrastive learning to regression is interesting" (reviewer ggUN). To our knowledge, SupCR is the only method centered on regression (We have thoroughly compared with it and provided analysis in our submission even if it was an Arxiv before submission), yet it faces a significant limitation: the necessity for data augmentation, which may not be suitable for modalities like time series or tabular data. SupReMix stands out as the first supervised contrastive learning framework for regression that does **NOT rely on data augmentation** ("This is significant" from reviewer gvot). Furthermore, earlier strategies such as MoCHi and i-Mix were implemented in a self-supervised manner, not in **supervised** training. Our approach utilizes regression labels to foster ordinality and continuity, aspects that were overlooked in previous methods.
>
> **(2) Technical Novelty:**
> The prior methods in contrastive learning and hard sample mining, including all those you've mentioned like SupCon, SupCR, Decoupled CL, MoCHi, and i-Mix, primarily focus on identifying hard negatives, leaving the concept of **hard positives** largely unexplored. We are grateful for your recognition of our "various interesting empirical studies of regression tasks, including visualizations of logit distribution and latent space." These studies underline the importance of incorporating hard positives in regression analysis. Our approach introduces the novel hard positive pairs in regression by combining two negative samples in such a way that their convex combination of labels matches the anchor's label. It aims to promote local linearity. SupReMix represents a groundbreaking approach in addressing the issue of hard positives in the regression context and proposing a solution leading to superior regression performance. Even in the broader scope of contrastive learning in general, hard positives remain an underexplored area. Wu et al. 2023, as cited in our related work, appears to be the only study delving into this concept, albeit in a self-supervised learning framework. On the other hand, SupReMix is the first contrastive learning framework that incorporates **label distance** into contrastive loss, providing a more fundamental approach to handle negative pairs, and it is supported by the concept of Distance Magnifying we proposed and theoretically analyzed.
>
> **(3) Theoretical Contribution:**
> SupReMix is supported by two theorems and a lemma we established, offering novel insights in supervised contrastive learning for regression. We demonstrate that by assigning weights to contrastive pairs based on label distance, our approach amplifies the penalty for negative pairs that are more distant relative to nearer ones. Besides, we establish that the SupReMix loss function has a tight bound, and it guarantees the ordinality and continuity of representations as it approaches the infimum.

---

> > ### Comment · Reviewer_4GQV · 2023-11-21
> > **Further Questions on Novelty**
> >
> > Thanks for the detailed response on the novelty issue. I agree with the first point (1) that the proposed SupReMix tackles the problem of contrastive learning for regression. However, the used techniques have some overlap with the existing methods, and should be compared in detail: Firstly, the concept of hard sample mining in contrastive learning is widely discussed. The focused problem of hard sample mining in regression can utilize the existing techniques and concepts (e.g., ideas, loss functions, and theoretical analysis), which limits the novelty of this paper. As the authors mentioned, similar works like [1] should be well discussed in the Introduction and Related Works sections to ensure novelty. Secondly, the proposed **label distance** to select local samples to mixup is similar to C-Mixup [2], which should be further discussed (or I cannot see the real technical contributions of SupReMix). From my perspective, selecting similar samples to mix is a commonly studied improvement direction for mixup augmentations on regression tasks (there are several under-review works in ICLR'2024 also taking this idea). Therefore, I will keep my score unchanged if this issue is not well explained. Besides, please provide a new list of the cited works in the comment windows for the convenience of reference, or it will be confusing and inconvenient to find the cited work.
> >
> > ### Reference
> >
> > [1] Synthetic Data Can Also Teach: Synthesizing Effective Data for Unsupervised Visual Representation Learning. AAAI, 2023.
> >
> > [2] C-Mixup: Improving Generalization in Regression. NeurIPS, 2022.

---

> ### Author Response · Authors · 2023-11-16
> **Response to Reviewer 4GQV [2/2]**
>
> **Comparisons with related work (W2):** We appreciate the important comment. We firstly would like to highlight that our SupReMix is a novel approach for $\textbf{representation learning}$ for regression tasks that stands apart from traditional $\textbf{regression learning}$ methods like C-Mixup[4]. As recommended, we conducted two new experiments to validate the effectiveness of SupReMix: (1) we evaluated the regression performance of SupReMix (with pre-training followed by a linear probe) in comparison to methods (both training-from-scratch and pretrain + linear probe) including MoCHi[2], i-Mix[3], C-Mixup[4], Vanilla Mixup, and M-Mixup[6]. SupReMix has achieved better performance than **ALL** of them using AgeDB dataset (see the top two parts of the table below); (2) we demonstrated that integrating SupReMix with C-Mixup can further enhance C-Mixup's performance (bottom part of the table below).
>
> | Metrics | MAE ↓ | MSE ↓ | GM ↓ |
> | ------- | ----- | ----- | ---- |
> | **Regression Learning (Train from scratch)** | | | |
> | Vanilla | 7.77 | 101.60 | 5.05 |
> | Mixup | 8.03 | 119.51 | 5.78 |
> | M-Mixup | 8.02 | 104.33 | 5.34 |
> | C-Mixup | 7.66 | 99.82 | 4.82 |
> | **Representation Learning (Linear Probe)** | | | |
> | MoCHi | 9.07 | 132.50 | 5.93 |
> | i-Mix | 8.50 | 117.96 | 5.52 |
> | SupCon | 8.23 | 113.91 | 5.31 |
> | SupCR | 7.86 | 102.84 | 5.15 |
> | **SupReMix (ours)** | **7.12** | **87.65** | **4.58** |
> | **SupReMix + Regression Learning** | | | |
> | SupReMix + C-Mixup | 7.17 | 90.21 | 4.79 |
>
> We also performed additional performance comparisons between SupReMix with C-Mixup[4] across four different datasets (see the table below). Here we show that SupReMix can **consistently outperform C-Mixup** train from scratch and **bring improvement for C-Mixup** with the SupReMix pretrain (bottom row of the table below).
>
> | Metrics (MAE ↓) | AgeDB | IMDB | HCP | UKB |
> | --------------- | ----- | ---- | --- | --- |
> | C-Mixup | 7.66 | 5.43 | 9.07 | 3.00 |
> | **SupReMix** | **7.12** | 5.38 | 8.43 | 2.97 |
> | **SupReMix + C-Mixup** | 7.17 | **5.21** | **8.23** | **2.90** |
> | GAINS (**Joint** VS. C-Mixup (%)) | **+6.4** | **+4.1** | **+9.3** | **+3.3** |
>
>
> **Hyper-parameters (W3):**
> We appreciate this useful point. To clarify, SupReMix introduces **ONLY ONE** additional hyper-parameter: window size (denoted as $\gamma$) for Mix-pos. This parameter is chosen based on the range of regression targets. Table 10 demonstrates that $\gamma$ is arguably robust: moderate ranges of $\gamma$ consistently enhance performance. Another consideration may be related to the beta distribution for Mix-neg. In Table 11, by experimenting with different shapes of distribution on six datasets with different modalities for different tasks, beta(2,8) consistently outperforms other variants. It indicates that as long as the level of difficulty is maintained at a certain level, i.e., the beta distribution leans towards the anchor, the algorithm can perform the task robustly.
>
> **Related Work (W4):**
> Thank you for pointing out these useful references. As recommended, we have included the following sentences to highlight these studies and clarify the research gaps. (In Introduction, second paragraph).
>
> "... Nevertheless, they mainly focused on hard negatives with hard positives underexplored, and the hardness of contrastive pairs in contrastive learning for regression remains inadequately examined. Data mixing techniques (Zhang et al., 2018; Verma et al., 2019; Shen et al., 2022), commonly used for data augmentation, have been used to create hard samples in previous contrastive learning methods for classification (Kalantidis et al., 2020; Lee et al., 2020; Liu et al., 2023). However, these methods do not leverage the label distance to differentiate the hardness among hard negative mixtures. ..."

---

> > ### Author Response · Authors · 2023-11-19
> > **Request for Feedback on Our Response**
> >
> > Dear Reviewer,
> >
> > we would greatly appreciate your feedback on whether our response adequately addressed your questions and concerns.

---

> > ### Comment · Reviewer_4GQV · 2023-11-21
> > **Further Questions on Experiments**
> >
> > Thanks to the authors for the detailed feedback and conducted experiments. The weaknesses W3 and W4 have been well solved, and I suggest the authors further polish the writing based on these issues. As for W2, I suggest the authors update these added results in the main text because these comparison experiments are important to show the effectiveness of SupReMix over existing methods. Moreover, I suggest the authors add the finetuning evaluation results as RNC [1], which is a widely used evaluation for self-supervised pre-training. Could the authors add some generative pre-training baselines (e.g., masked image modeling [2] or masked language modeling [3]) with the finetuning evaluation? I interesting in
> >
> > ### Reference
> >
> > [1] Rank-N-Contrast: Learning Continuous Representations for Regression. NeurIPS, 2023.
> >
> > [2] Masked Autoencoders Are Scalable Vision Learners. CVPR, 2022.
> >
> > [3] RoBERTa: A Robustly Optimized BERT Pretraining Approach. ArXiv, 2019.

---

> > > ### Author Response · Authors · 2023-11-22
> > > **discussion window close in less than one day, looking forward to your reply**
> > >
> > > Dear Reviewer 4GQV,
> > >
> > > As the discussion period ends in **less than one day**, we kindly request to know if our responses have addressed all your concerns. If so, could you please consider increasing your score? We are more than happy to provide further clarification if time allows.

---

> ### Author Response · Authors · 2023-11-21
> **Further Clarification on Novelty**
>
> Thank you so much for your reply! We would like to further emphasize the novelty of our method. We believe that a clearer articulation of novelty will distinctly highlight our contributions.
>
> We agree that the concept of hard-sample mining (e.g., MoChi[6]) and the usage of label distance for mixup (e.g., C-Mixup[7]) have been studied in previous work. We would like to underscore the following key points:
>
> 1. SupReMix's Contribution on hard-sample mining: Previous hard-sample mining strategies typically **“fail to address hard positives” as they cannot be applied to generate hard positives**. In contrast, SupReMix introduces a unified framework that effectively generates both hard positive and negative samples. This is accomplished through our innovative anchor-inclusive and anchor-exclusive mixup strategy, integrating both ordinality and continuity into the process. To the best of our knowledge, the most relevant work, Wu et al [5] is the only one addressing hard positives. However, it is **limited to image data**, and the generation of hard positives requires training a GAN model which is known to be **difficult to optimize**.
>
> 2. Comparison of Our Approach with C-MixUp in Terms of Label Distance Usage: Unlike C-MixUp, which selects samples based on label distance (with all samples potentially being selected), our method drastically differs by **leveraging label distance differently for positives and negatives**.
>    - For negatives, label distances serve as weights for **controlling the degree of strength of repulsion**. This fundamental difference in approach allows for a more nuanced treatment of negatives, beyond mere sample selection.
>    - For positives, unlike C-MixUp, which gives all potential pairs a certain probability, SupReMix uses a defining window size $\gamma$, strongly dependent on the local linearity assumption. Therefore, we do not consider samples with a large label distance (even with low probability) for positives.
>
> 3. Differentiating Our Approach with Concurrent Work (We understand the concerns raised, and while we believe it is not necessary to address this point, we are more than happy to further highlight our contribution): Concurrently, we recognize similar efforts in using label distance for regression in other works. Notably, we are the **first to theoretically analyze a property of contrastive regression relevant to label distance**, which we have termed Distance Magnifying (DM). By applying label distance as a weight for negatives, we enhance the penalty for more distant negative pairs compared to nearer ones. Additionally, our loss function demonstrates a tight bound, ensuring the ordinality and continuity of representations as it converges to the infimum.

---

> ### Author Response · Authors · 2023-11-21
> **Additional Experiments**
>
> We thank the reviewer for acknowledging that we have solved W2, W3, W4 well. For W2, as recommended, we have added comparisons with C-Mixup[7], RoBERTa[4] into our main text. For the rest compared methods, we will add complete comparison for all six datasets for the camera-ready version.
>
> ## Finetuning
>
> Noted that RNC[1] (“SupCR” previously) is an Arxiv paper when submitting, and we have already compared it with all six datasets following their linear probe scheme (**please note that the original RNC[1] paper only provides linear-probe results**). However, we thank reviewers for suggesting comparing SupReMix with RNC[1] under the setting of fine-tuning. As recommended, we here provide evaluation with fine-tuning for SupCon[2]/RNC[1]/SupReMix on AgeDB and IMDB dataset. We observe that SupReMix consistently outperforms SupCon[2] and RNC[1] under the fine-tuning evaluation.
>
> | Dataset | AgeDB (MAE) | IMDB (MAE) |
> |---------|-------------|------------|
> | SupCon[2] | 8.83 | 6.46 |
> | RNC[1] | 8.12 | 6.07 |
> | SupReMix | 7.23 | 5.39 |
>
> ## Comparing with generative pre-training baselines
>
> We thank the reviewer for the interest in our method. We compare our method with MAE[3] and RoBERTa[4] for with linear-probe on AgeDB and STS-B respectively. SupReMix consistently outperforms these methods.
>
> | Metric | MAE | MSE | GM |
> |--------|-----|-----|----|
> | MAE (ViT-B) | 11.23 | 192.53 | 7.46 |
> | MAE (ViT-L) | 10.88 | 183.25 | 7.29 |
> | SupReMix (ResNet-50) | 7.12 | 87.65 | 4.58 |
>
> |     | rho | MSE | MAE |
> |-----|-----|-----|-----|
> | RoBERTa | 0.751 | 0.987 | 0.777 |
> | SupReMix (BiLSTM) | 0.760 | 0.893 | 0.759 |
>
> [1] Zha, Kaiwen, et al. "Rank-N-Contrast: Learning Continuous Representations for Regression." NIPS 2023.
>
> [2] Khosla, Prannay, et al. "Supervised contrastive learning." NIPS 2020.
>
> [3] He, Kaiming, et al. "Masked autoencoders are scalable vision learners." CVPR 2022.
>
> [4] RoBERTa: A Robustly Optimized BERT Pretraining Approach. ArXiv, 2019.
>
> [5] Synthetic Data Can Also Teach: Synthesizing Effective Data for Unsupervised Visual Representation Learning. AAAI, 2023.
>
> [6] Hard Negative Mixing for Contrastive Learning. In NeurIPS, 2020.
>
> [7] C-Mixup: Improving Generalization in Regression. In NeurIPS, 2022.

---

> > ### Comment · Reviewer_4GQV · 2023-11-22
> > **Feedback to Additional Experiments and Clarification on Novelty**
> >
> > Thanks for the extensive finetune experiments and the clarification on novelty. I appreciate the analysis and experiments in this work, and I think my concerns about experiments and implementation details (W2~W4) were nearly tackled. The additional experiments can be included in the main text to make the evaluation more diverse. However, as for the novelty, I am not convinced by the further clarification, and I decided to maintain my rating of 5. I suggest the authors add these clarifications in the main text. Moreover, I just found a recently published work [1] on augmentations for regression tasks, which can be included in the related work section.
> >
> > ### Reference
> > [1] Anchor Data Augmentation. NeurIPS, 2023.

---

> > > ### Author Response · Authors · 2023-11-22
> > > **Appreaciate your review and effort !**
> > >
> > > Thank you for your feedback and suggestions! We are glad to learn that your concern regarding implementation and experiment has been resolved. We've added the additional experiments (including generative pre-training baseline and fine-tuning evaluation result) to Appendix D.9 and D.10 and adjusted our introduction based on our previous clarification about novelty. This very recent work[1] has been incorporated into our related work section. Thank you again for helping us improve our paper!
> > >
> > > [1] Anchor Data Augmentation. NeurIPS, 2023.

---

### Author Response · Authors · 2023-11-22
**Response to All Reviewers**

We thank all the reviewers for acknowledging the contributions of our work and providing insightful comments and suggestions! We are glad to see that the reviewers found that:
1. We are tackling an important and well-motivated problem (Reviewer 4GQV, gvot, DdrH), which traditionally receives less attention.
2. The formulation of our method is intuitive (Reviewer gvot, DdrH), interesting (Reviewer ggUN), and clever (Reviewer gvot).
3. Our approach doesn’t rely on data augmentation, which is significant (Reviewer gvot).
4. Our proposed method shows superior, encouraging performance across a quite wide range of applications (Reviewer gvot, DdrH).
5. Our experiment provides interesting empirical studies (Reviewer 4GQV).
6. Our presentation is excellent (Reviewer ggUN). The manuscript is well-organized, clearly written, and easy to follow, which provides empirical analysis and theoretic explanations. (Reviewer 4GQV).

We have made the following changes in our work and updated the manuscript (major changes are highlighted in blue in the updated version), and we believe that we have addressed all concerns raised by reviewers:
1. We have incorporated the writing, math notation, figure, and reference suggestions in the updated paper (Sections 1, 2, 3.2, 3.3). We have modified the title of the paper to make it more descriptive.
2. We have added experiments in comparison with more regression learning baselines for each dataset. And we show that our proposed SupReMix is compatible with them and can further enhance their performance (Appendices D.5, D.6, and Tables 1-6), as suggested by Reviewers 4GQV, gvot, ggUN.
3. We have added experiments on multi-dimensional targets and comparison with baseline methods (Appendix D.7), as suggested by Reviewers gvot, DdrH.
4. We have added a comparison with generative pre-training methods for image and text datasets (Appendix D.9), as suggested by Reviewer 4GQV.
5. We have added experiments in comparison regarding different “pair selection strategies” for mixup (Appendix D.7), as mentioned by Reviewer gvot.

We thank all reviewers for providing constructive feedback, which we believe has truly improved our paper. We are more than happy to offer further clarifications.

---

### Meta-Review · Area_Chair_UTae · 2023-12-23

**Metareview:**

The paper addresses representation learning for regression and introduces Supervised Contrastive Learning for Regression with Mixup (SupReMix). The proposed approach leverages anchor-inclusive and anchor-exclusive mixtures as hard negative and positive pairs, fostering richer ordinal information in the embedding space. Through extensive experiments on various regression datasets and theoretical analysis, the paper demonstrates that SupReMix pre-training enhances the learning of continuous ordered representations, leading to improvements in regression performance.

The reviewers are somewhat split about this paper. Reviewers ggUN and 4GQV have raised several issues regarding experiments (methodology, comparisons), somewhat incremental improvements, and concerns regarding novelty, but the other two reviewers are more positive. There is also the concern that more revisions are needed at this time to make the paper palatable; and new experiments and insights would need another round of reviews to evaluate the overall effectiveness and the claims in this work.

After extended deliberation in the end the decision was made to reject the paper. Ultimately, the weaknesses in the evaluation and the somewhat incremental contribution were just too hard to overlook. We hope the reviews are useful for improving and revising the paper.

**Justification For Why Not Higher Score:**

Same as above.

**Justification For Why Not Lower Score:**

N/A

---

### Decision · Program_Chairs · 2024-01-16

Reject